# Membrane lipid nanodomains modulate HCN pacemaker channels in nociceptor DRG neurons

Lucas J. Handlin [1], Natalie L. Macchi[1], Nicolas L. A. Dumaire [2], Lyuba Salih[2], Erin N. Lessie[1], Kyle S. McCommis [1], Aubin Moutal [2] & Gucan Dai [1] ✉

Cell membranes consist of heterogeneous lipid nanodomains that influence key cellular processes. Using FRET-based fluorescent assays and fluorescence lifetime imaging microscopy (FLIM), we find that the dimension of cholesterol-enriched ordered membrane domains (OMD) varies considerably, depending on specific cell types. Particularly, nociceptor dorsal root ganglion (DRG) neurons exhibit large OMDs. Disruption of OMDs potentiated action potential firing in nociceptor DRG neurons and facilitated the opening of native hyperpolarization-activated cyclic nucleotide-gated (HCN) pacemaker channels. This increased neuronal firing is partially due to an increased open probability and altered gating kinetics of HCN channels. The gating effect on HCN channels is likely due to a direct modulation of their voltage sensors by OMDs. In animal models of neuropathic pain, we observe reduced OMD size and a loss of HCN channel localization within OMDs. Additionally, cholesterol supplementation inhibited HCN channels and reduced neuronal hyperexcitability in pain models. These findings suggest that disturbances in lipid nanodomains play a critical role in regulating HCN channels within nociceptor DRG neurons, influencing pain modulation.

Membrane lipids form distinct compartments known as lipid domains, which serve as specialized regions within cell membranes for localized cell signaling. In compartmentalized membranes, a distinctive type known as detergent-resistant lipid-ordered membrane domain (OMD) stands apart from other regions, such as the disordered membrane domains. This distinction arises from the composition of OMDs, which is characterized by a heightened presence of cholesterol and sphingolipids[1–3]. The OMDs possess an intriguing variation in size, ranging from a few nanometers to hundreds of nanometers. This dynamic nature allows the lipid domains to undergo rapid changes, coalescing or splitting apart, in response to different physiological conditions. Techniques such as super-resolution microscopy, single-molecule fluorescence, and cryo-electron tomography have been used to visualize and characterize these membrane domains[4–7]. Nevertheless, many details about the structure, function, and significance of OMDs remain elusive.

The significance of lipid domains has been recognized in diverse biological processes[1,8]. For instance, in Alzheimer's disease, the dysregulation of lipid domains has been implicated in the accumulation of amyloid-β plaques[9–11]. In viral infections, these domains can serve as entry sites for the viruses into cells[12,13]. Inflammatory responses involve the reorganization of lipid domains, influencing cell signaling and immune responses. Moreover, emerging evidence suggests a link between lipid domains and chronic pain, particularly neuropathic pain, which is often caused by nerve injuries[14,15]. In an animal model of chemotherapy-induced peripheral neuropathy and arthritis, targeting lipid domains was effective in alleviating the pathological symptoms[16]. However, the connection between OMDs and neuropathic pain remains understudied. Manipulating these domains may hold

[1]Edward A. Doisy Department of Biochemistry and Molecular Biology, Saint Louis University School of Medicine, Saint Louis, USA. [2]Department of Pharmacology and Physiology, Saint Louis University School of Medicine, Saint Louis, USA. ✉e-mail: gucan.dai@health.slu.edu

potential for developing targeted treatments for diseases related to lipid domain dysfunction, including neurodegenerative disorders, viral infections, inflammatory conditions, and neuropathic pain.

Pacemaker hyperpolarization-activated cyclic nucleotide-gated (HCN) channels are critical in several important physiological processes[17]. These channels are involved in establishing the resting membrane potential (RMP) and regulating the frequency of action potentials, thereby contributing to the automaticity of excitable cells[17]. Moreover, HCN channels are important for pain sensation, as evidenced by their involvement in the repetitive firing of primary nociceptor neurons in the dorsal root ganglion[18–20]. HCN2-channel current is responsible for eliciting repetitive firing in nociceptor neurons[21,22]. In support of this, knocking out the HCN2 gene reduces neuropathic pain in animal models[18,23]. Conversely, increased HCN channel activity correlates with spontaneous action potential firing and chronic pain, while pharmacological blockade of HCN channel activity decreases inflammatory and neuropathic pain[18,20,24]. The lipid domain-specific localization of HCN channels appears to significantly influence their function, thereby impacting neuronal and cardiac excitability[25–27]. The function of cardiac HCN4 channels depends on compartmentalized cAMP signaling in lipid microdomains[28,29]. We showed that the voltage-sensing process of HCN channels, including the voltage-sensor movement, is directly regulated by OMDs[30]. Additionally, the voltage-sensing modules of HCN channels interact intimately with membrane lipids, including signaling phosphoinositides[31–34]. In summary, these studies corroborate a significant connection between pain sensation and the regulation of pacemaker HCN channels by membrane lipids.

In this current study, we perform fluorescence lifetime imaging microscopy (FLIM) and Förster resonance energy transfer (FRET), which are well-suited for evaluating the nanometer sized OMDs in various cell types. Nociceptor DRG neurons exhibit larger OMDs than non-nociceptive DRG neurons. When OMDs are disrupted in nociceptor DRG neurons, the action potential firing becomes more frequent. HCN2 channels, abundantly expressed in nociceptor DRG neurons, tend to localize within the OMD and are sensitive to the disruption of lipid domains. Furthermore, the voltage sensor movement of HCN2 channels is modulated directly by the OMD localization. This modulation appears to impact the gating of the native HCN channels in nociceptor DRG neurons. In addition, significant reductions in the OMD size are associated with both a chemotherapy-induced and a nerve injury-induced model of neuropathic pain. These findings suggest that disruptions in lipid domains, leading to increased HCN channel open probability and accelerated channel activation, may contribute to the mechanisms underlying neuropathic pain.

## Results

### Imaging the OMDs of living cells using FLIM-FRET
We employed cholera toxin subunit B (CTxB)-conjugated FRET donors and acceptors as probes to investigate lipid nanodomains. CTxB, derived from *Vibrio cholerae*, acts as a pentameric multivalent ligand with exceptionally high affinity (picomolar) for GM1 gangliosides[35,36]. These gangliosides, specifically GM1, are distinct sphingolipid molecules prominently present in OMDs. Conventional fluorescence microscopy encounters limitations due to the diffraction limit of light (250 nm), often making it impossible to visualize small-sized lipid domains. FRET-based fluorescence microscopy can overcome this limitation by acting as a spectroscopic distance ruler with nanometer sensitivity. This sensitivity is ideal for detecting nanoscale lipid domains of the dimensions considered critical in biological membranes[37]. We used Alexa Fluor 488 (AF-488) CTxB conjugates as FRET donors and Alexa Fluor 647 (AF-647) CTxB conjugates serving as FRET acceptors (Fig. 1a). This specific FRET pair was previously established, with an Förster critical distance ($R_0$) value of ~52 Å[38]. When the size of lipid domains exceeds a radius of ~10 nm, the likelihood of accommodating two CTxB pentamers within the same lipid domain

increases significantly[36]. In contrast, when the lipid domain has a radius of less than 5 nm, it can accommodate only one CTxB pentamer[36], resulting in minimal FRET between AF-488 and AF-647. This approach can effectively quantify the relative size of lipid domains.

To conduct live-cell time-resolved FRET imaging, we used a confocal scanning FLIM system to measure FRET efficiency (see Methods). We labeled the plasma membrane of tsA-201 cells—a modified version of human embryonic kidney (HEK) cells — by supplementing the cell culture medium with 20 nM CTxB AF-488 conjugates (the FRET donor). By using a pulsed laser for excitation, the frequency-domain lifetime information corresponding to each image pixel in the intensity-based image was transformed into a pixel on a Cartesian coordinate system, known as "phasor plot," which exhibits a semicircular shape (Fig. 1b and Supplementary Fig. 1)[39]. This semicircle serves as a reference for pure single-exponential lifetimes. In contrast, multiexponential lifetimes are observed within the boundaries of the semicircle. Notably, shorter lifetimes demonstrate a clockwise movement along the semicircle. The phasor approach offers several advantages, including an unbiased measurement of fluorescence lifetimes without the necessity of exponential fitting, and the capability to map multiple cursors to identify distinct lifetime species within confocal images[39–42] (Supplementary Fig. 1). In the subsequent phasor analysis of FRET, we specifically focused on the lifetime species originating from membrane-specific fluorescence while excluding those from the internalized CTxB AF-488 due to endocytosis (see Supplementary Fig. 1 and Methods).

When both FRET donors (CTxB AF-488) and acceptors (CTxB AF-647) were present at a concentration of 20 nM, the lifetime characteristics observed in labeled tsA cells exhibited a clockwise shift towards shorter lifetimes, indicating the occurrence of FRET (Fig. 1c). To calculate the FRET efficiency, we used the FRET trajectory method enabled by the phasor plot. The FRET trajectory begins at the zero FRET efficiency position, aligning with the donor-alone cursor, and proceeds through the measured phasor species. Ultimately, it reaches the theoretical 100% FRET efficiency located at the origin coordinate (0, 0). At this point, the background noise primarily consists of white noise with an infinite lifetime contribution. Moreover, the FRET efficiency for each pixel was determined using the FRET trajectory function, generating a heatmap based on FRET efficiency, to provide an overview of the FRET distribution of a single cell (Fig. 1d). The observed FRET values in tsA cells were low, implying a potential scenario in which numerous OMDs had small nanometer sizes and bound only a single CTxB[36]. This observed size range aligns with the estimated dimensions of lipid domains derived from the correlation length of the critical fluctuations in plasma membrane vesicles[43,44].

We further investigated the impact of the cholesterol-extracting reagent β-cyclodextrin (β-CD) and amphipathic ionic detergents on FLIM-FRET efficiency in tsA-201 cells. Our findings indicate that the acute (~1–2 min) application of β-CD (5 mM) significantly reduced FRET, while supplementing cholesterols using 1 mg/mL preloaded methyl-β-CD-cholesterol complexes "water soluble cholesterol" (WSC) elevated FRET. In addition, the treatment of micromolar concentrations of ionic detergents, such as sodium dodecyl sulfate (SDS) and Triton X-100, increased FRET levels (Fig. 1 d–g). Amphipathic detergents are regarded as "cholesterol-phobic," indicating their tendency to compete with cholesterol in disordered domains. This competition promotes cholesterols to migrate towards larger OMDs. These results suggest that the extraction of cholesterols leads to separation of OMDs, the addition of cholesterol expands OMDs, whereas low concentrations of detergents (50 μM SDS or 30 μM Triton X-100) produce the merging of OMDs. Previous research has demonstrated that application of amphipathic detergents triggers clathrin-independent massive endocytosis (MEND) during the application of Triton X-100 or after the application of SDS[3,45]. MEND was able to internalize a substantial portion of the plasma membrane[3,45,46]. Our results lend support

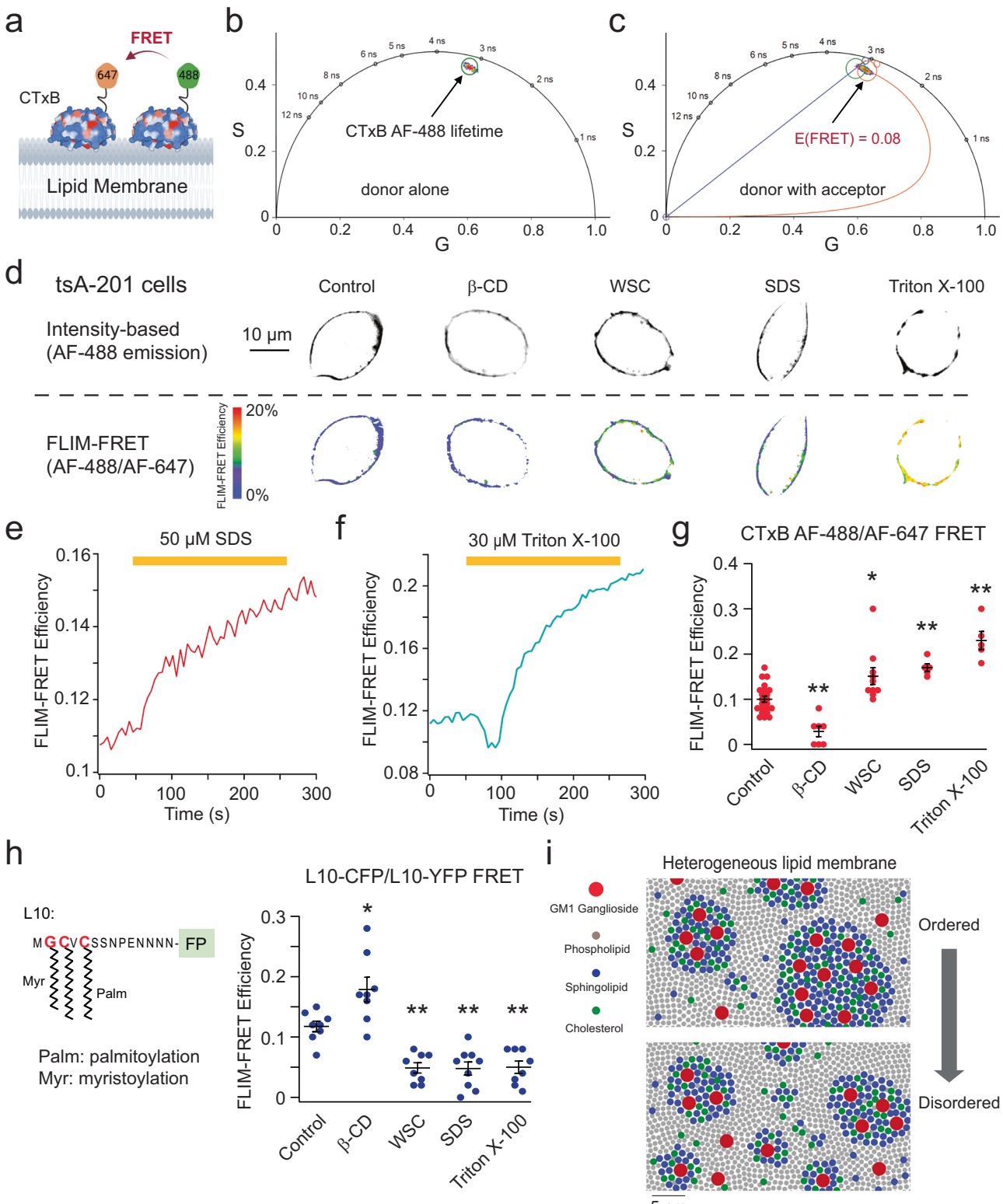

to the notion that detergent-induced MEND involves the coalescence of OMDs.

In addition, we employed a genetically encoded fluorescent marker of the OMD, L10, derived from the initial 10 amino acids of the Lck kinase[30,47,48], to reinforce our conclusions (Fig. 1h). The membrane-localized L10 probe, palmitoylated at two cysteines, exhibits colocalization with the OMDs in a GM1-independent manner. Using FLIM-FRET between L10-CFP and L10-YFP pairs, similar to the CTxB-based method, enabled us to monitor the OMD size. Consistent with prior research

using this probe[48], treatment with β-CD increased FRET, while WSC treatment decreased FRET between L10-CFP and L10-YFP (Fig. 1g and Supplementary Fig. 2a–f). Note that the directional change in FRET here opposes that observed with CTxB-based FRET, as the L10 probes come closer in distance when the OMD is smaller. In contrast, a smaller OMD size accommodates fewer CTxB molecules, often just one, leading to lower FRET (Fig. 1i). Therefore, while the CTxB and L10-based FRET pairs produce opposite FRET changes, the results imply consistent conclusions regarding OMD size. The FRET efficiency significantly decreased

**Fig. 1 | Assessing the OMD size using FLIM-FRET and FRET-based OMD probes.**
**a** Experimental design using the FRET donor CTxB AF-488 with the FRET acceptor CTxB AF-647. **b, c** Representative phasor plots of membrane-localized fluorescence of tsA201 cells from donor alone (CTxB AF-488) or with CTxB AF-647. The FRET efficiency was calculated using the FRET trajectory (in red). **d** Representative intensity-based and FLIM-FRET based heatmap images illustrate the effects of 5 mM β-CD, 1 mg/mL WSC, 50 µM SDS and 30 µM Triton X-100 on the plasma membrane of tsA cells. Time course of the increase of the FRET efficiency after applying 50 µM SDS (**e**) or 30 µM Triton X-100 (**f**). **g** Summary of the effects produced by the acute application of 5 mM β-CD ($n = 7$ cells, $**p = 0.001$), 1 mg/mL WSC ($n = 10$ cells, $*p = 0.01$), 50 µM SDS ($n = 5$ cells, $**p = 0.006$) and 30 µM Triton X-100 ($n = 5$ cells, $**p = 3e-7$) as shown in panel (**e**) and (**f**), compared to the control ($n = 22$ cells). Data

shown are mean ± s.e.m., $*p < 0.05$; $**p < 0.01$, using one-way ANOVA (no adjustment). **h** Diagram of the L10 probe with N-terminal fluorescent protein; summary of the effects produced by the acute application of 5 mM β-CD ($n = 8$ cells, $*p = 0.012$), 1 mg/mL WSC ($n = 8$ cells, $**p = 0.004$), 50 µM SDS ($n = 9$ cells, $**p = 0.002$) and 30 µM Triton X-100 ($n = 8$ cells, $**p = 0.005$) compared to the control ($n = 8$ cells). Data shown are mean + s.e.m., one-way ANOVA (no adjustment) was used. **i** Illustration showing ordered and disordered lipid membrane domains from an extracellular view, highlighting the distribution of GM1 gangliosides, phospholipids, sphingolipids, and cholesterol. Smaller OMDs can only host a single CTxB, whereas larger OMDs have the capacity to bind multiple CTxB molecules. The scale bar corresponds to the estimated diameter of 7 Å for the area of a single phospholipid. The lipid composition may not accurately represent its native state.

with low concentrations of detergents in the case of L10 probes, affirming OMD coalescence under these conditions (Fig. 1h). Furthermore, using another similarly engineered marker, S15, which tends to colocalize with disordered membrane regions, revealed a contrasting change in the S15-CFP/S15-YFP FRET after β-CD treatment (Supplementary Fig. 3). Finally, to test any potential influence from the membrane topology and accessibility, we pre-incubated the cells with a hypotonic solution (206 mOsm) to induce membrane swelling. The results using the L10-based FLIM-FRET pairs showed that the directional changes in FRET-reported OMD sizes were not significantly impacted by this treatment (Supplementary Fig. 2g). Collectively, these results serve to further validate our approach using CTxB-based FRET pairs.

### Domain size of OMDs varies depending on the cell type
Using FLIM-FRET and CTxB Alexa Fluor 488 or 647 conjugates, we conducted measurements to determine FRET efficiency indicative of OMD size in various primary cells and cultured cell lines (Fig. 2). These included isolated rat dorsal root ganglion (DRG) somatosensory neurons, isolated mouse primary hepatocytes, isolated mouse dermal fibroblasts, isolated mouse cardiac pacemaker cells, cultured embryonic rat cardiomyocytes (H9c2 cells), and tsA-201 cells. Importantly, significant differences were observed between the different cell types, with pacemaker cells and small-diameter DRG neurons demonstrating highest FRET levels (Fig. 2a–d). Hepatocytes, known for their cholesterol synthesis, also showed high FRET, indicative of large OMDs. Conversely, fibroblasts, embryonic kidney cells, and cultured H9c2 cardiomyocytes exhibited low FRET. These results document substantial heterogeneity in lipid domain sizes in the plasma membrane of different cell types.

DRG neurons, which are pseudo-unipolar sensory neurons, play a crucial role in transmitting sensory information from the periphery to the spinal cord, particularly for pain sensation[49,50]. DRG neurons that have a relatively small diameter, and mostly unmyelinated axons (C-fibers) are responsible for relaying nociceptive, thermal, and mechanoreceptive signals[51]. Conversely, large-diameter DRG neurons, which are non-nociceptive, mostly transmit mechanoreceptive and proprioceptive signals through large-diameter myelinated axons[51–53]. We found that the CTxB-based FLIM-FRET efficiency in the smaller neurons with cell capacitance less than ~28 picofarad (pF) was significantly greater than that in neurons of larger size (Fig. 2d). To further substantiate this finding, we extended the application of L10 probe-based FRET pairs to primary cells, including small and large DRG neurons as well as hepatocytes. In line with the observed larger OMD size in small DRG neurons and hepatocytes, the L10 FRET pairs exhibited reduced FRET efficiencies in these cell types in comparison to large DRG neurons and tsA cells (Fig. 2d). These results suggest that heterogeneity in the OMD size of sensory neurons could potentially affect the excitability of these neurons.

### Lipid tail structure influences the OMD size in an in vitro fatty liver model
The formation of OMDs relies on the tight packing of sphingolipids with saturated lipid tails and cholesterols. To investigate the impact of

supplementing fatty acids such as palmitate and oleate on the FRET between CTxB-conjugated AF-488 and AF-647, we implemented protocols commonly used to generate in vitro models of fatty liver disease. For hepatocytes, the overnight treatment of additional 300 µM palmitate in the culture medium in the presence of albumin resulted in a significant increase in FRET efficiency (Supplementary Fig. 4a, b). However, combining 200 µM palmitate with 100 µM oleate nearly abolished the FRET signal, indicating a division of the OMDs (Supplementary Fig. 4a, b).

In this cell model of fatty liver, the addition of palmitate induces cellular stress on the cells, potentially leading to lipotoxicity[54,55]. Conversely, the inclusion of oleate can act as a preventive measure against such cellular stress[54]. Notably, oleate is a monounsaturated fatty acid (18:1) with a double bond that induces a tilt and increased flexibility in the carbon chain, in contrast to the saturated aliphatic chain (16:0) of palmitate. Introducing the tilted aliphatic chain to the lipid membrane greatly impedes the formation of OMDs. These results suggest that the lipotoxicity introduced by palmitate is associated with a substantial increase in the abundance of OMDs, and oleate supplementation can rescue this effect. Moreover, these findings further validate the reliability of our approach using the CTxB-based probes in assessing the size and abundance of OMDs.

### Disrupting OMDs accelerates action potential firing of nociceptor DRG neurons
Following the validation experiments using FLIM-FRET to measure OMDs in various cell types, we shifted our focus to DRG sensory neurons. To investigate the potential effects of disrupting OMDs on the firing behavior of small nociceptor DRG neurons, we conducted whole-cell current clamp experiments. Specifically, we tested whether altering the size of these domains has an impact on the generation of action potentials. To achieve this, we used an acute application of β-CD (5 mM) to disrupt OMDs, as confirmed by the observed reduction in FRET efficiency using the CTxB FRET pairs following its application (Fig. 3a). OMD disruption was also confirmed by an increase in L10 probe-based FRET following β-CD treatment (Fig. 3b). In contrast, acute 1 mg/mL WSC treatment did not further increase the FRET efficiency, likely because the OMD dimension was already high in small DRG neurons (Fig. 3a, b).

We found that the β-CD treatment increased the spontaneous action potential firing activity of nociceptor DRG neurons (Fig. 3c, d). We supplemented our cell culture with nerve growth factor (NGF), which promoted the spontaneous firing of DRG neurons[56], mimicking a moderate pain condition. Furthermore, with positive current injections, the action potential-firing frequency increased as expected and showed adaptive behavior in some cells (Fig. 3e). Cells treated with β-CD showed more frequent current injection-elicited firing than control cells (Fig. 3e, Supplementary Fig. 5a, b). The observed augmentation in the spontaneous and the elicited firings of these DRG neurons by the disruption of OMDs suggests that these neurons exhibit a state of hyperactivity. This finding is consistent with a pain phenotype, further suggesting that the alteration of OMDs contributes to the modulation of neuronal excitability during the development of chronic pain. We

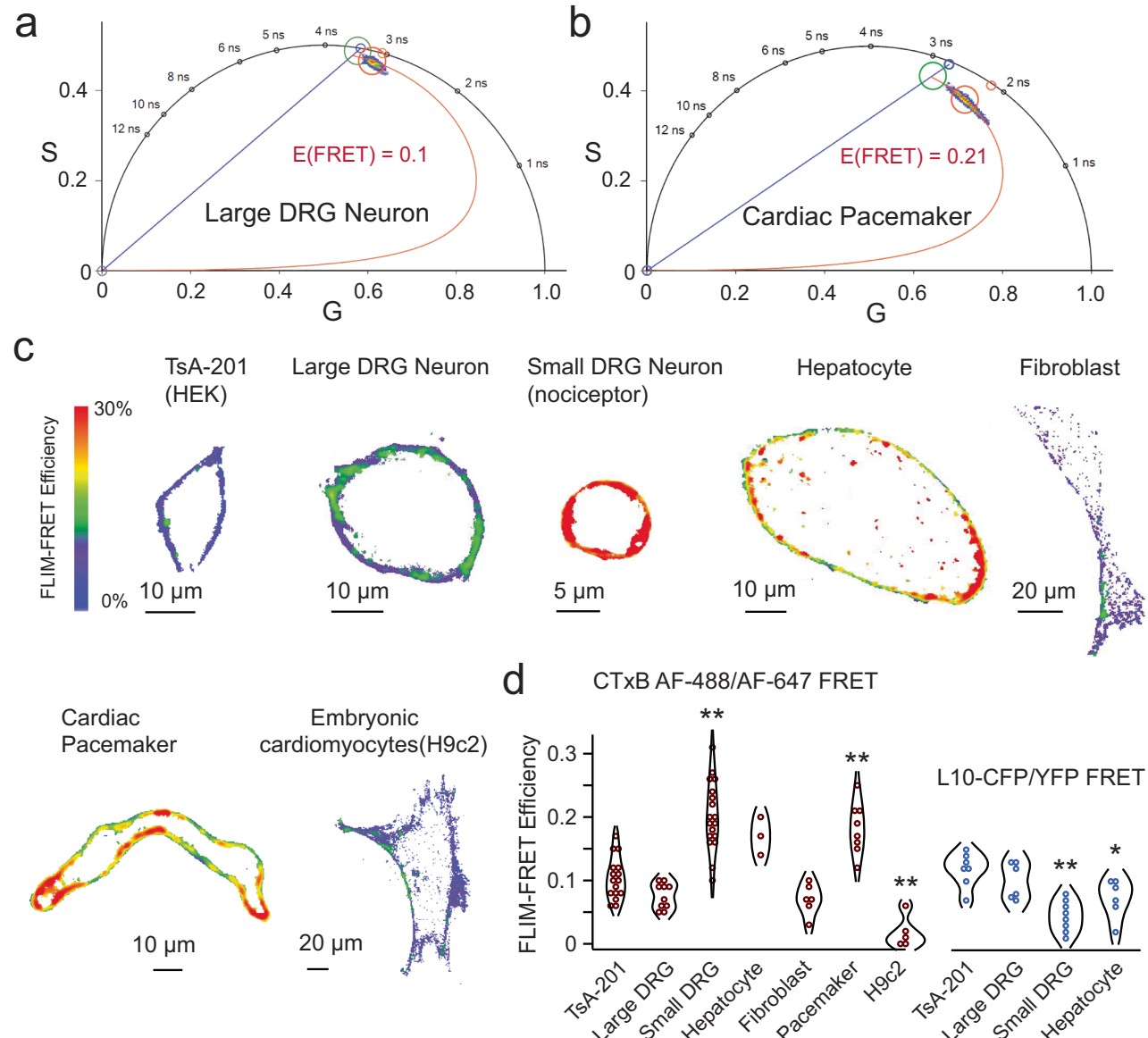

**Fig. 2 | Live-cell FLIM-FRET reports the relative OMD dimension in different cell types. a** Representative phasor plot of a large DRG neuron showing the CTxB AF-488/CTxB AF-647 FLIM-FRET. Using the FRET trajectory, the average FRET efficiency is estimated at 0.1. **b** Representative phasor plot of a cardiac pacemaker cell. Using the same FRET pair and FRET trajectory method, the average FRET efficiency of this cell is estimated at 0.3. **c** Representative single-cell heatmap images of the membrane-localized FLIM-FRET efficiencies of different cell types as illustrated. **d** Summary data of the FRET efficiencies of different cell types tested in panel (**c**) using the CTxB-based FRET pairs: $n = 16$ cells for tsA cells, as compared to other cell types using one-way ANOVA (no adjustment); $n = 10$ cells for large DRG neurons, $p = 0.6$; $n = 18$ cells for the small DRG neurons, $**p = 1e-8$; $n = 3$ cells for hepatocytes, $p = 0.1$; $n = 6$ cells for fibroblasts, $p = 0.6$; $n = 8$ cells for cardiac pacemakers, $**p = 3e-4$; $n = 5$ cells for H9c2 cells, $**p = 0.001$. The right side of the panel is the summary data of comparable experiments using transiently transfected L10-based FRET pairs: $n = 7$ cells for tsA cells; $n = 6$ cells for the large DRG neurons, $p = 0.7$; $n = 8$ cells for the small DRG neurons, $**p = 2e-4$; $n = 6$ cells for hepatocytes, $*p = 0.03$. Data shown are mean ± s.e.m., $*p < 0.05$; $**p < 0.01$.

also tested the addition of cholesterol using the treatment of 1 mg/mL WSC. In the presence of NGF, the percentage of spontaneous firing reduced from 82% after the β-CD treatment ($n = 17$) to 40% for the control ($n = 15$) and 30% after the WSC treatment ($n = 10$). The ability of nociceptor DRG neurons to fire action potentials in response to positive current injections was slightly impaired after WSC treatment, with a more significant difference observed compared to the β-CD condition than the control in pairwise comparisons (Fig. 3d, e, Supplementary Fig. 5c). These results suggest that expanding the OMD by cholesterol enrichment could be used to rescue membrane hyperexcitability and potentially alleviate pain.

Considering the role of oleate supplementation in disrupting membrane order, we applied an overnight treatment of 100 μM bovine

serum albumin (BSA)-conjugated oleate to DRG neurons. We found that compared to the BSA alone control treatment, the oleate treatment mainly potentiated the current injection-elicited action potential firing of small DRG neurons (Supplementary Fig. 6a). These results suggest that oleic acids may play a role in regulating sensory neurons through perturbing OMDs of the membrane.

**Paclitaxel, a drug that produces chemotherapy-induced neuropathy, reduces the OMD size**

Chemotherapy-induced peripheral neuropathy (CIPN) affects ~40% of chemotherapy patients and lacks effective treatments at present[57]. Paclitaxel (Taxol), a widely used chemotherapeutic agent, is known to induce both acute and chronic CIPN. Considering the role of OMDs in

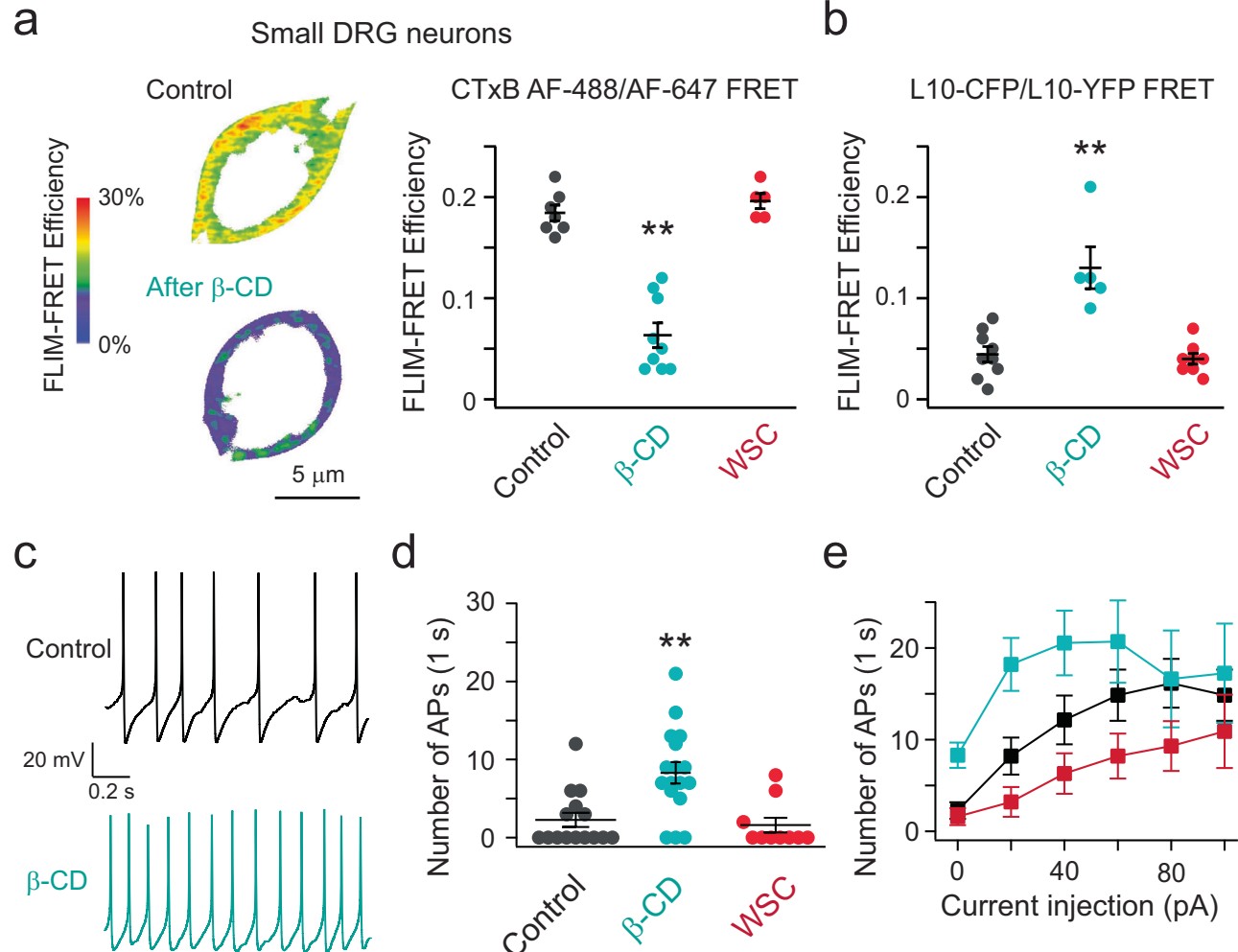

**Fig. 3 | Disrupting the OMD accelerates the action potential firing of DRG neurons. a** Representative heatmap FRET images of small nociceptor DRG neurons before and after the acute β-cyclodextrin treatment. Summary data of the effects of β-cyclodextrin ($n = 9$ cells, **$p = $3e-7) and WSC ($n = 5$ cells) on the FRET efficiency between CTxB 488/647 probes. Data shown are mean ± s.e.m., $n = 7$ for the control. One-way ANOVA (no adjustment) was used. **b** Summary data of the effect of β-CD ($n = 5$ cells, **$p = $6e-5) and WSC ($n = 8$ cells) on the FRET efficiency between the L10-CFP and L10-YFP pair. Data shown are mean ± s.e.m., $n = 9$ cells for the control. One-way ANOVA (no adjustment) was used. **c** Representative spontaneous action potential firings of small DRG neurons before and after the acute β-CD treatment.

**d** Summary data of the effects of β-CD ($n = 17$ cells, **$p = 0.001$) and WSC ($n = 10$ cells) on the number of the un-elicited action potential firings during the 1 s recording period, compared to the control ($n = 15$ cells). Data shown are mean ± s.e.m. One-way ANOVA (no adjustment) was used. **e** Summary data of current injection-elicited action potential firing frequency. Same data with 0 pA current injection and the same number of cells as in panel (**d**); **$p = 0.009$ with 20 pA current injection between the control and the β-CD conditions; **$p = 0.004$ with 40 pA current injection between the β-CD and WSC conditions; *$p = 0.04$ with 60 pA current injection between the β-CD and WSC conditions. Data shown are mean ± s.e.m., one-way ANOVA was used.

regulating voltage-gated ion channels and the previously reported role of paclitaxel in regulating membrane lipid domains[14,30,58,59], we postulated that the paclitaxel-triggered neuropathy might be associated with alterations in the lipid domain organization of nociceptor DRG neurons. Using FLIM-FRET and the same CTxB-mediated labeling approach, our results revealed that overnight treatment using either a low concentration (10 nM) or a saturating concentration (1 μM) of paclitaxel resulted in a significant reduction in the FLIM-FRET efficiency of small DRG neurons (Fig. 4a–d). In contrast, the same treatment of paclitaxel did not produce a considerable reduction in the FLIM-FRET efficiency of large DRG neurons, which could be because the smaller OMDs in large DRG neurons are more resistant to this treatment (Fig. 4d). Additionally, the increased FLIM-FRET efficiency observed with L10-CFP/L10-YFP-based probes further confirms the reduction in OMD size following paclitaxel treatment (Fig. 4e). Moreover, consistent with the results seen with the β-CD treatment, the action potential firing of nociceptor DRG neurons accelerated after the

treatment of 10 nM paclitaxel (Fig. 4f, g). This finding implies that the disruption of OMDs is associated with and may play a direct role in paclitaxel-induced neuropathy. Since paclitaxel targets tubulin and microtubules, this finding may suggest a linkage between OMDs and cellular cytoskeleton.

### Facilitated native HCN currents of nociceptor DRG neurons after OMD disruption

Pacemaker hyperpolarization-activated HCN channels are key players in regulating the electrical excitability of DRG neurons[18,22]. With physiological electrochemical driving force, they conduct inward currents primarily carried by sodium ions, depolarizing the membrane potential and bringing it closer to the threshold of firing an action potential. Small nociceptor DRG neurons display relatively slow HCN currents that are highly sensitive to cAMP, while larger DRG neurons exhibit faster HCN currents with reduced sensitivity to cAMP[18,22]. The native HCN channel currents of small nociceptor DRG neurons were elicited

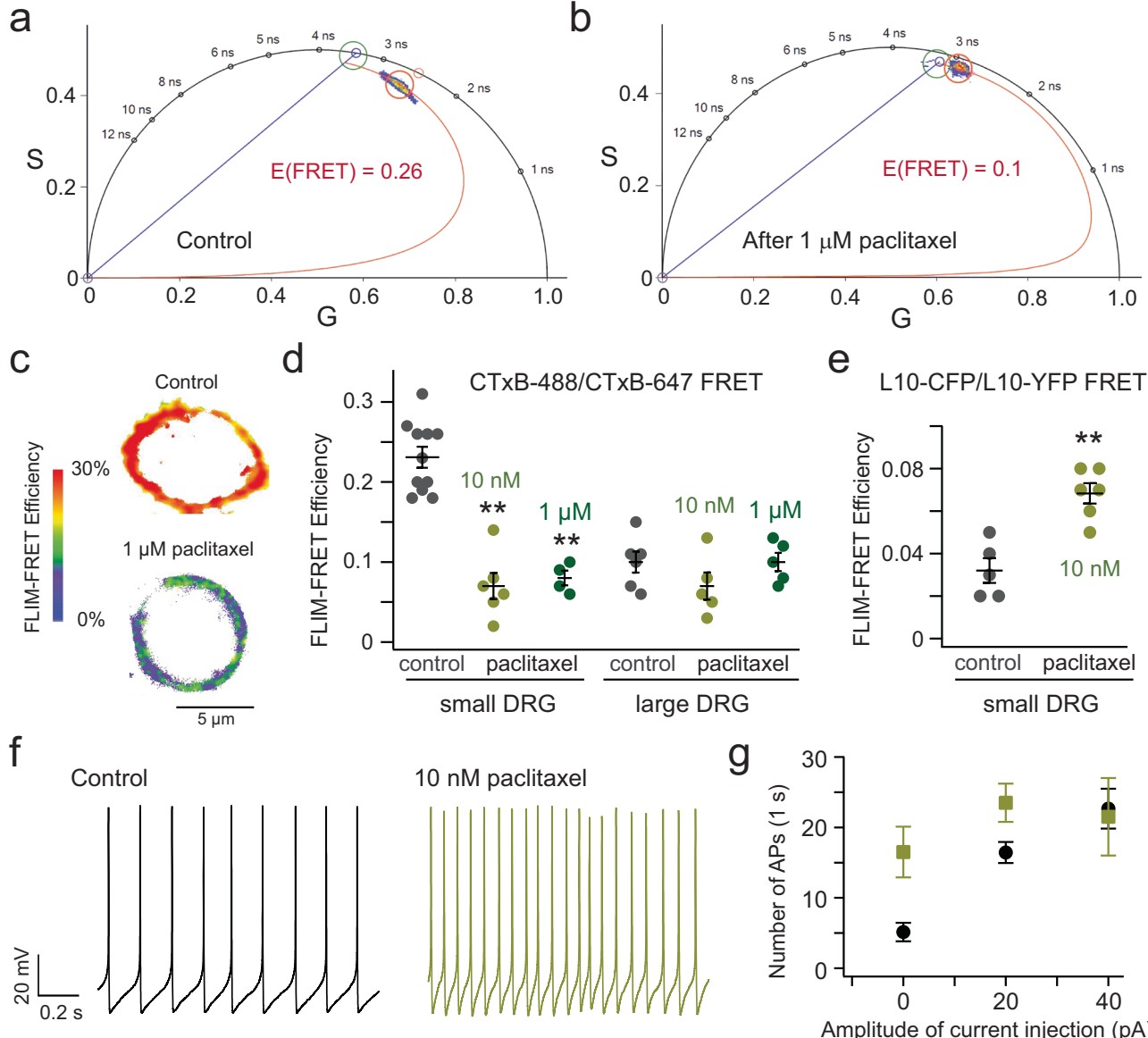

**Fig. 4 | Effects of the CIPN-related drug paclitaxel on the OMD size and excitability of DRG neurons. a, b** Representative phasor plots of a small nociceptor DRG neuron before and after the overnight 1 μM paclitaxel treatment, using the CTxB-AF 488/CTxB-AF 647 FLIM-FRET. **c** Representative heatmap images of membrane-localized FLIM-FRET efficiencies of small nociceptor DRG neurons before and after the paclitaxel treatment. **d** Summary data of the effect on the CTxB-based FRET efficiencies of small and large DRG neurons caused by overnight 10 nM or 1 μM paclitaxel treatments. Data shown are mean ± s.e.m., $n = 11$ cells for the control, $n = 6$ cells with 10 nM paclitaxel (**$p = 7e$-7), $n = 4$ cells with 1 μM paclitaxel (**$p = 1e$-5), for small DRG neurons. For large DRG neurons, $n = 6$ cells for the control, $n = 5$ cells for both 10 nM paclitaxel and 1 μM paclitaxel conditions.

One-way ANOVA (no adjustment) was used. **e** Summary data of the effect on the L10 probe-based FRET efficiencies of small DRG neurons caused by the 10 nM paclitaxel treatments. Data shown are mean ± s.e.m., $n = 5$ cells for the control and n = 6 with paclitaxel, **$p = 9e$-4. Two-sided student's $t$-test was used. **f** Representative spontaneous action potential firings of small DRG neurons before and after the overnight 10 nM paclitaxel treatment. **g** Summary data results of the number of action potentials of small DRG neurons plotted against the amplitude of current injection. Data are shown as mean ± s.e.m. $n = 7$ cells for the control and 4 cells with paclitaxel treatment, **$p = 0.006$ with 0 pA current injection, *$p = 0.03$ with 20 pA current injection. Two-sided student's $t$-test was used.

by a series of hyperpolarizing voltages using whole-cell voltage clamp, exhibiting a conductance-voltage (G-V) relationship with a voltage for the half-maximal activation ($V_{1/2}$) of −93 mV, close to that of HCN2 channels (Fig. 5a)[18,22]. Adding a supersaturating concentration of cyclic adenosine monophosphate (cAMP, 0.5 mM) in the pipette solution shifted the G-V relationship to more depolarized voltages, with a $V_{1/2}$ of −83 mV (Fig. 5d, k, i), consistent with that HCN2 channels are sensitive to cAMP. Contrastingly, larger-diameter DRG neurons abundantly express the HCN1 isotype, which is not particularly sensitive to cAMP[22]. The change in the $V_{1/2}$ ($\Delta V_{1/2}$), by 0.5 mM cAMP for the G-V relationship of endogenous HCN channels within large DRG neurons, is around

5 mV (Supplementary Fig. 7a, b), only half of that observed in small DRG neurons.

We found that disrupting OMDs using the same acute application of β-CD altered the hyperpolarization-dependent activation of endogenous HCN channels of small nociceptor DRG neurons. First, we investigated the kinetics of channel gating. Endogenous HCN currents in small DRG neurons exhibited double-exponential kinetics, with the first component ($\tau_1$) representing the primary time constant. With β-CD treatment, channel activation accelerated, with $\tau_1$ reduced from 82 ms in the control to 64 ms on average (using −140 mV hyperpolarization, Fig. 5g). The second time constant ($\tau_2$) for the channel

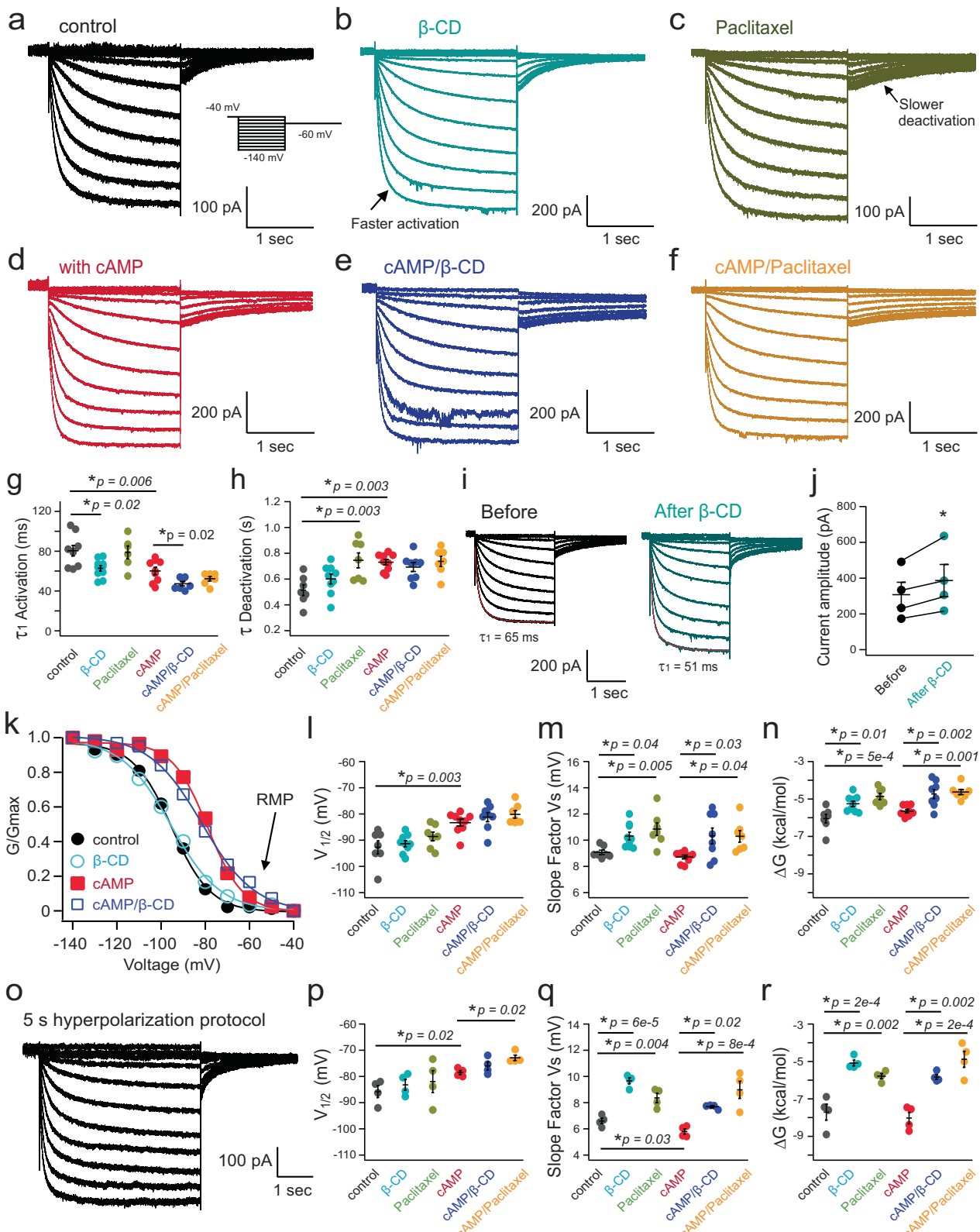

activation was not significantly altered with β-CD treatment. The deactivation of endogenous HCN currents, which was predominantly mono-exponential, remained largely unaffected by β-CD (Fig. 5h). In the presence of cAMP, a similar acceleration after β-CD in channel activation was observed (Fig. 5g). This faster channel activation likely underlies the enhanced neuronal firing associated with β-CD treatment.

Then, we examined the equilibrium properties of channel gating. The HCN current measurements described above were collected from different individual cells, so the maximum current amplitude was influenced by factors such as the number of channels expressed and the proportional expression of channel isotypes. To assess whether β-CD alters the channel open probability, we applied 2.5 mM β-CD to the same patch and monitored the change in current amplitude before and

**Fig. 5 | Impact of OMD disruption on the gating of endogenous HCN channels in nociceptor DRG neurons. a–f** Representative HCN currents elicited by hyperpolarizing voltages. In panel (**b**), the activation time constants are $\tau_1 = 59$ ms, $\tau_2 = 388$ ms, compared to $\tau_1 = 79$ ms, $\tau_2 = 399$ ms in panel (**a**), at −140 mV. The deactivation time constant is 869 ms in panel (**c**), compared to 448 ms in panel (**a**). **g, h** Summary data of the effects of β-CD, 10 nM paclitaxel, and cAMP on the HCN kinetic gating parameters with saturating hyperpolarizing voltages: $\tau_1$ of channel activation and τ of channel deactivation. Data shown are mean ± s.e.m., $n = 9$ patches for the control, $n = 10$ for β-CD, $n = 6$ for paclitaxel conditions in the absence of added cAMP; and n = 9 patches for the control, $n = 7$ for β-CD, $n = 7$ for paclitaxel conditions in the presence of 0.5 mM cAMP. One-way ANOVA (no adjustment) was used. **i** Representative endogenous HCN currents of small DRG neurons before and after β-CD for the same patch. **j** Summary of the same patch experiments as in panel

(**i**). Data shown are mean ± s.e.m., $n = 4$, *$p = 0.04$, two-sided paired *t*-test. **k** Representative G-V relationships of the HCN currents in the following conditions: control, with β-CD, with cAMP, and with both β-CD and cAMP. **l–n** Summary of the effects of β-CD, 10 nM paclitaxel, and cAMP on parameters: $V_{1/2}$, slope factor, and the free energy for the channel activation. Data shown are mean ± s.e.m., $n = 7$ patches for the control, $n = 9$ for β-CD, $n = 7$ for paclitaxel conditions in the absence of added cAMP; and $n = 9$ patches for the control, $n = 8$ for β-CD, $n = 7$ for paclitaxel conditions with 0.5 mM cAMP, *$p < 0.05$; **$p < 0.01$ using one-way ANOVA (no adjustment). **o** HCN currents elicited by longer hyperpolarizing voltage pulses (5 s), without cAMP. **p–r** Summary of effects of β-CD, paclitaxel, and cAMP on HCN gating parameters under longer voltage protocols. Data shown are mean ± s.e.m., $n = 4$ patches, one-way ANOVA (no adjustment).

after treatment. This approach revealed a 25.7 ± 3.4% increase in current amplitude within ~5 min after the β-CD application, indicating a significant potentiation of the channel's open probability (Fig. 5i, j). In control experiments, α-CD, which cannot efficiently extract cholesterols and affect OMDs[48], failed to increase the current amplitude during the same-patch experiment, only decreasing the current amplitude by 22.9 ± 4.0%. This decrease, or "current run-down," was also observed in untreated controls (17.1 ± 7.0% reduction in current amplitude) and was likely due to a gradual loss of membrane phosphoinositides after establishing the whole-cell patch-clamp recording mode[34,60]. Following this increase in peak current amplitude by β-CD observed from the same-patch experiments, we normalized the elicited HCN currents recorded from different cells (control versus β-CD preincubation) to their respective peak amplitudes. We then analyzed the voltage dependence of HCN gating by plotting the G-V relationship. Without cAMP supplementation, β-CD treatment did not significantly shift the $V_{1/2}$. Instead, it increased the slope factor of the G-V relationship according to the Boltzmann sigmoidal fit, suggesting less gating charge movement during the channel activation (Fig. 5 k, l, and Supplementary Table 1). This increase in the slope factor was less pronounced in the control experiments using α-CD application (Supplementary Table 1). Furthermore, with 0.5 mM cAMP in the pipette solution, β-CD treatment also failed to shift the $V_{1/2}$ for the channel activation to more depolarized voltages (Fig. 5 k, l). Nevertheless, the slope factor associated with the Boltzmann fit of the G-V relationship increased significantly, suggesting that in the presence of cAMP, the estimated gating charge movement across the membrane was decreased (Fig. 5k, m, and Supplementary Table 1). This change in the effective gating charge movement could be related to the membrane-thinning effect after disrupting the OMD, which might narrow the focused electric field of the membrane. On the other hand, the shallower slope of the G-V relationship produces a higher channel open probability at the RMP (around −60 mV). Specifically, the relative open probability of HCN channels, as reflected in the normalized G-V relationship and quantified at −60 mV and −70 mV, also increased after the β-CD treatment, both with and without cAMP (Supplementary Fig. 8). Collectively, the observed increases in open probability of HCN channels could effectively depolarize the membrane to promote action potential firing. Finally, using the obtained parameters from the Boltzmann fits, the free energy (Δ*G*) associated with the HCN channel activation can be calculated. A lower amount of free energy was required to open the channel after the β-CD treatment, regardless of the presence of cAMP (Fig. 5n and Supplementary Table 1).

We also tested a prolonged hyperpolarizing voltage protocol (5 s, Fig. 5o) to activate HCN channels in small DRG neurons, ensuring more stable channel activation before analyzing instantaneous tail currents. In comparison, the shorter voltage protocol (2 s) emphasized the kinetic aspect of the channel activation. Consistent with our expectations, β-CD produced similar effects. The extended protocol reduced the cAMP-induced shift in the G-V relationship while enhancing the impact of β-CD on the slope factor and the free energy

change (Fig. 5 p–r). In the absence of the additional cAMP, the estimated apparent charge movement during channel activation was 3.9 in the control group and 2.6 with β-CD. With 0.5 mM cAMP supplementation, these values increased to 4.3 in the control group and 3.3 with β-CD (also see Methods). These observations suggest that disruption of the OMD led to accelerated channel activation, elevated open probability, reduced voltage-sensing charge movement across the membrane electric field, and energetic facilitation of the HCN channel opening process.

Next, we evaluated the effect of overnight 10 nM paclitaxel on the gating of HCN channels in small DRG neurons using both the short and long hyperpolarizing voltage protocols (Fig. 5). Regarding channel gating kinetics, paclitaxel did not accelerate channel activation but instead slowed channel deactivation. This effect on gating kinetics is distinct from that of β-CD, suggesting that factors other than OMDs contribute to the effects of paclitaxel (Fig. 5g, h). The slowed channel deactivation indicates a more stabilized channel opening, which could contribute to the increased neuronal firing observed in CIPN. After paclitaxel treatment, the HCN G-V relationship exhibited a slight shift in the $V_{1/2}$, also differing somewhat from the effect observed with β-CD, particularly in the presence of 0.5 mM cAMP. Like the effect of β-CD, the slope became significantly shallower (Fig. 5 i–n, p–r). After the paclitaxel treatment, the estimated apparent charge movement during channel activation was reduced to 3.1 without the added cAMP and 2.9 with 0.5 mM cAMP, compared to controls, as determined using the long hyperpolarizing voltage protocol (Fig. 5p–r). The ΔG associated with the HCN channel activation was also decreased considerably after the paclitaxel treatment (Fig. 5n, r). On the contrary, the administration of 10 nM paclitaxel did not produce a significant effect on the gating parameters of HCN channels in large DRG neurons (Supplementary Fig. 7c, d). These findings suggest that the paclitaxel-induced changes in HCN channel gating, specific to small DRG neurons, likely contribute to the hyperexcitability observed in DRG neurons following this type of chemotherapy. Moreover, the overnight oleate treatment similarly decreased the apparent gating charge movement for native HCN channels in small DRG neurons, suggesting a similar mechanism of disrupting OMDs underlying the oleate-induced neuronal hyperexcitability (Supplementary Fig. 6b–d).

## OMD disruption impacts the domain localization and voltage sensor conformations of HCN2 channels

We also tested whether the primary HCN-channel type HCN2 channels expressed in nociceptor DRG neurons are localized in OMDs. Using the phasor FLIM-FRET method and the OMD-specific probe L10-CFP, we found considerable FRET between L10-CFP and a human HCN2-YFP construct expressed in tsA 201 cells (Supplementary Fig. 9). L10-CFP was shown to have low FRET with human HCN1-YFP but exhibited higher FRET with the human HCN4-YFP[30]. Functionally, similar to the effects observed with endogenous HCN channels in DRG neurons, acute treatment with 5 mM β-CD did not change the $V_{1/2}$ but increased the slope factor of the G-V relationship for overexpressed hHCN2

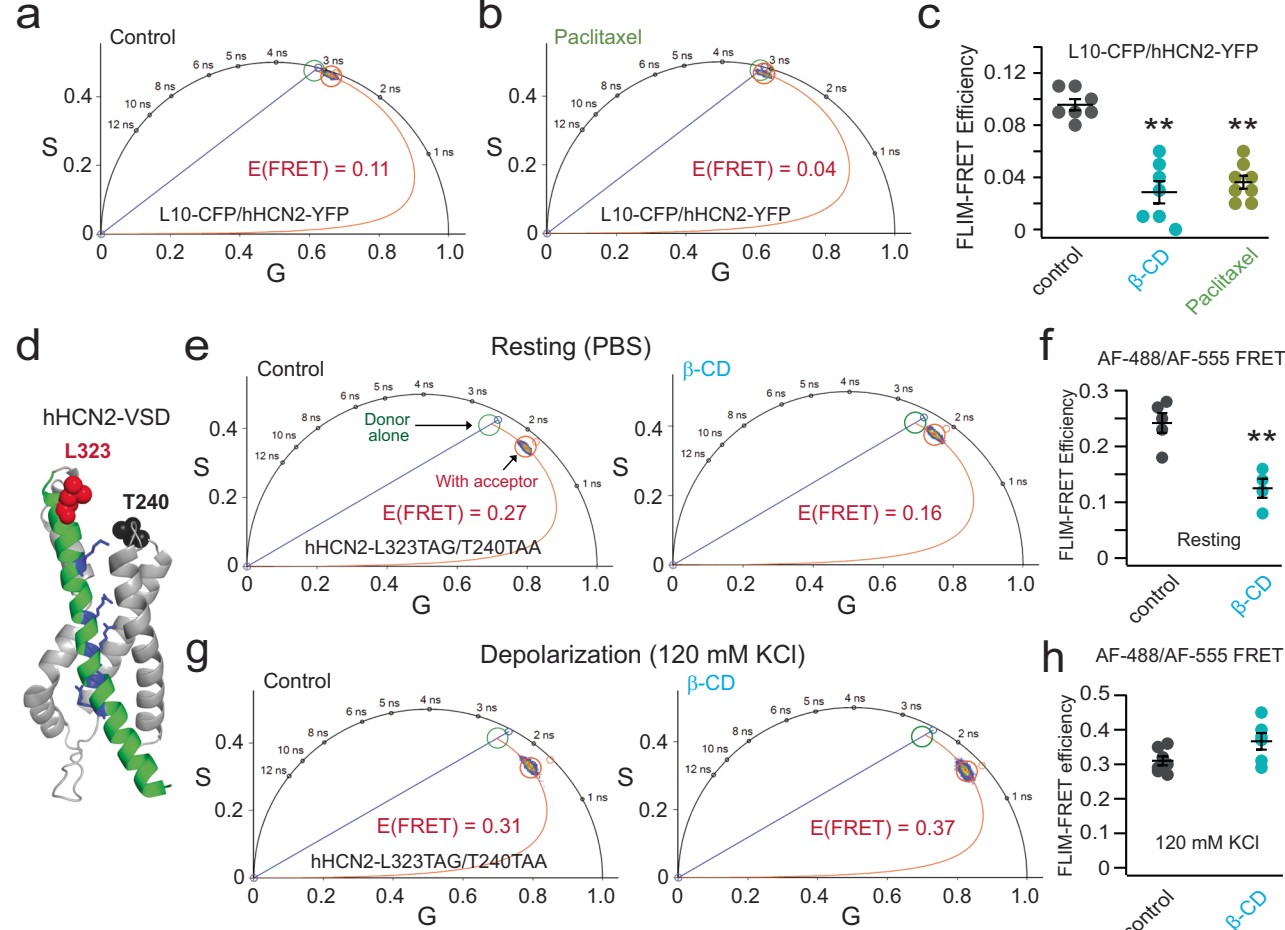

**Fig. 6 | Phasor FLIM-FRET measurements of hHCN2 channel localization and voltage sensor rearrangements. a, b** Representative phasor plots showing FRET between L10-CFP and hHCN2-YFP expressed in small DRG neurons before and after 10 nM paclitaxel treatment. **c** Summary data of the effects of the β-CD and paclitaxel on the FRET between L10-CFP and hHCN2-YFP. Data shown are mean ± s.e.m., n = 7 for control and β-CD conditions, n = 8 for paclitaxel conditions, **p = 1e-6 for β-CD and **p = 4e-6 for paclitaxel compared with the control. One-way ANOVA (no adjustment) was used. **d** Cartoon showing the design of the FRET sites in the VSD of human HCN2 channels and the representative phasor plot showing FRET between donor AF-488 labeled to the HCN2-L323TAG site and the FRET acceptor AF-555

labeled to the HCN2-T240TAA site. **e** Representative phasor plot of the same FRET pair as in panel (**d**) following acute 5 mM β-cyclodextrin treatment. **f** Summary data of the effect of the β-CD on the FRET efficiency of hHCN2-L323TAG/T240TAA channels. Data shown are mean ± s.e.m., n = 5 cells for the control and n = 4 cells after β-CD, **p = 0.002 using two-sided student's t-test. **g** Representative phasor plot of the same FRET pair as in panels (**d**) and (**e**), following acute 5 mM β-cyclodextrin treatment with 120 mM KCl in the bath. **h** Summary data of the effect of the β-CD on the FRET efficiency as shown in panel (**g**). Data shown are mean ± s.e.m., n = 7 cells for the control and n = 6 cells with β-CD, p = 0.054 using two-sided student's t-test.

channels in tsA-201 cells (Supplementary Fig. 10). The free energy for channel activation of hHCN2 was intermediate between that of hHCN1 and hHCN4 (Supplementary Fig. 10a), with β-CD facilitating channel opening by reducing the free energy required for channel activation (Supplementary Fig. 10).

When transiently transfected into small DRG neurons, coexpression of L10-CFP probes and HCN2-YFP channels resulted in significant FRET (Fig. 6a, c). These results indicate that, like human HCN4 channels, human HCN2 channels tend to localize to OMDs. Additionally, acute treatment with 5 mM β-CD and overnight exposure to 10 nM paclitaxel led to a substantial reduction in FRET between L10-CFP and HCN2-YFP (Fig. 6b, c). Consistent with the OMD localization of hHCN2, the FRET between S15-CFP and HCN2-YFP was significantly lower than that observed with L10-CFP, tested in tsA cells (Supplementary Fig. 9).

We tested whether the voltage sensor S4 helix of human HCN2 undergoes rearrangement following disruption of the OMD, similar to previous findings in human HCN4 channels[30]. The structural foundation of gating charge is established by positively charged arginines

within the S4 transmembrane helix, which traverse the electric field across the membrane. Employing a dual stop codon (amber TAG and ochre TAA)-mediated suppression strategy, we incorporated two noncanonical amino acids, trans-cyclooct-2-ene-L-lysine (TCO*K) and N-propargyl-L-lysine (ProK), using click chemistry for site-specific fluorescence labeling, as in prior studies[30,61]. This system uses two orthogonal archaea pyrrolysine-based stop-codon suppression strategies in tsA201 cells[61] (Supplementary Fig. 11a). For fluorescence labeling of these noncanonical amino acids, we employed the strain-promoted inverse electron-demand Diels–Alder (IEDDA) cycloaddition reaction to conjugate TCO*K with tetrazine-conjugated AF-488[61]. For the site of ProK, copper(I)-catalyzed azide-alkyne cycloaddition (CuAAC) was used to attach a picolyl azide-conjugated Alexa Fluor 555 (AF-555) (Supplementary Fig. 11b–d)[61]. The FRET pair for hHCN2 was established with L323TAG at the N-terminal side of the S4 helix and T240TAA in the S1-S2 helix loop of the voltage-sensing domain of the channel (Fig. 6d). Upon the acute application of 5 mM β-CD, we observed a significant decrease in FRET between AF-488 and AF-555 using the phasor FLIM-FRET method (Fig. 6e, f). Without the FRET

acceptor, the lifetime of the donor AF-488 alone was not changed by the β-CD application (Supplementary Fig. 11e). This reduction in FRET closely mirrored the changes in FRET between corresponding sites in hHCN4 channels[30]. In contrast, as shown previously, minimal FRET change was detected between comparable sites in hHCN1 channels following β-CD treatment[30]. When a high potassium (120 mM KCl) solution was used to depolarize the membrane, increased FRET between AF-488 and AF-555 was detected (Fig. 6g, h). However, the effect of β-CD treatment on the FRET efficiency was much attenuated with the membrane depolarization, suggesting a dependence of the OMD-mediated modulation on the functional state of the voltage sensor (Fig. 6g, h). This state dependence of FRET change by OMD disruption was previously observed in hHCN4 channels[30]. These findings suggest that the voltage sensor S4 helix of hHCN2 similarly undergoes rearrangements when the channel transitions from OMDs to disordered membrane regions. This may underlie the structural mechanism of the observed decrease in the apparent gating charge of HCN channels after disrupting the OMD in DRG neurons.

## Modulation of HCN channels by OMDs in a nerve injury-induced model of neuropathic pain

Mechanical allodynia and neuropathic pain are known to manifest following peripheral nerve damage. To define the potential correlation between nerve injury-induced neuropathic pain and changes in the size of OMDs, we also used the FLIM-FRET method to assess OMD size in DRG neurons in a spared nerve injury (SNI) model of neuropathic pain[62]. Animals showed expected pain phenotypes in our behavior tests (Fig. 7a). In the SNI model, neuronal firing was elevated in the ipsilateral side of the injury (Fig. 7b, and also see Fig. 8). We measured the percentage of spontaneous action potential firing of small nociceptor DRG neurons, which elevated from 56% ($n = 9$) on the contralateral side of the injury to 100% on the ipsilateral side ($n = 12$). Furthermore, our study revealed that the FRET efficiency, which reports OMD size, displayed a reduction in small DRG neurons due to the nerve injury (Fig. 7c). Specifically, the FRET efficiency measured from small DRG neurons decreased from $10.8 \pm 1.1\%$ for the control neurons on the contralateral side of the injury to $4.4 \pm 0.4\%$ for the neurons on the ipsilateral side of the injury. For large DRG neurons that are characterized by small OMDs, the FRET was also lower in the ipsilateral ones than in the contralateral ones, but the difference was less significant compared to small DRG neurons (Fig. 7c). These results suggest that downsizing the OMD was not only observed in the in vitro paclitaxel-induced neuropathy model but also in the in vivo nerve injury-induced pain model, implying that it might be a general phenomenon associated with neuropathic pain. Additionally, we coexpressed L10-CFP probes and hHCN2-YFP channels in small DRG neurons within the SNI model to assess the colocalization of HCN2 channels with the OMD. We observed pronounced FRET for the control neurons on the contralateral side. In contrast, this FRET was significantly lower for neurons on the ipsilateral side of the injury (Fig. 7d, e). These results indicate the disruption of OMDs in the SNI pain model, consequently leading to a predominant localization of HCN2 channels in disordered lipid domains.

We assessed the gating parameters of native HCN channels in neurons of the SNI pain model. Kinetically, the channel activation was accelerated while the deactivation was slowed (Fig. 7f, g). Again, the analysis of equilibrium properties included determining the $V_{1/2}$ for channel activation, the slope factor reporting apparent gating charges associated with channel activation, and the $\Delta G$, with and without cAMP, in neurons from both the contralateral and ipsilateral sides of the injury. Our findings aligned with observations from β-CD treated neurons and those from the paclitaxel-induced CIPN pain model. There was a significant reduction in the apparent gating charge on the ipsilateral side, accompanied by a noticeable shift in $V_{1/2}$ for channel activation toward more depolarized voltages compared with the contralateral side of the injury (Fig. 7h–I, Supplementary Table 1). The $\Delta V_{1/2}$ is ~6 mV, a more significant effect than comparable experiments with β-CD and paclitaxel. Consequently, the calculated $\Delta G$ was similarly diminished in neurons from the ipsilateral side (Fig. 7k). Furthermore, the differences in the slope factor of the G-V curve and $\Delta G$ for channel activation persisted with cAMP supplementation, although the $\Delta V_{1/2}$ between the contralateral and ipsilateral sides became indistinguishable (Fig. 7i–k, Supplementary Table 1). These results are consistent with a direct modulation of the voltage sensor of HCN channels by OMDs in SNI model-pain conditions.

## Dual effects of cholesterol on HCN channels in nociceptor DRG neurons

Given that cholesterol enrichment can potentially suppress neuronal firing, we applied acute WSC treatment to small nociceptor DRG neurons and measured the endogenous HCN currents (Fig. 8). This treatment resulted in a ~7 mV shift of the G-V relationship of channel activation towards more hyperpolarized voltages, indicating an inhibitory effect on HCN gating (Fig. 8a, b). Notably, there was no statistically significant change in the slope factor of the G-V relationship (Fig. 8c), which contrasts with the significant increase in the slope factor observed after the β-CD treatment to these neurons (Fig. 5). The free energy change due to WSC treatment was also minimal (Fig. 8d). This may be explained by our observation that WSC treatment did not further increase FRET efficiencies that report OMD dimensions (Fig. 3a, b). This effect of WSC on the $V_{1/2}$ but not the slope factor of the G-V relationship suggests a distinct mechanism, possibly involving direct interaction between cholesterol and HCN channels in nociceptor DRG neurons. While a putative cholesterol binding site has been proposed near the S4-S5 linker region of human HCN3 channels[63], it has not been validated in other HCN channels.

We tested whether cholesterol enrichment could ameliorate the pain phenotype observed in the SNI model. Given that the OMD size was significantly reduced in small DRG neurons on the ipsilateral side of the SNI, we hypothesized that WSC supplementation could restore the OMD in these neurons. We conducted CTxB-based FLIM-FRET experiments on ipsilateral small DRG neurons after WSC supplementation and observed an increased FRET signal at the plasma membrane. This indicates that WSC supplementation restored the OMD size to levels comparable to those in contralateral side nociceptor neurons (Fig. 8e). Additionally, neuronal firing in the ipsilateral small DRG neurons was restored after the WSC supplementation. Specifically, the percentage of spontaneous neuronal firing decreased to 25% ($n = 8$, Fig. 8f), and the number of action potentials induced by current injections was reduced (Fig. 8g). Interestingly, WSC treatment did not significantly shift the $V_{1/2}$ for HCN channel activation but decreased the slope factor of the G-V relationship (Fig. 8j). Consequently, the free energy required to stabilize channel closing was lower after WSC treatment (Fig. 8k). Consistent with this, the main effect of WSC supplementation on HCN gating kinetics in ipsilateral small DRG neurons was accelerated deactivation, indicating that the closed channel state became more favorable (Fig. 8l).

Overall, these findings suggest that cholesterol enrichment at the membrane has two distinct effects: increasing OMD size and directly binding to the channels. An increase in OMD size likely modifies membrane thickness and alters the movement of voltage sensors, reducing the apparent gating charge movement. Direct cholesterol binding may become functionally significant when the OMD size is sufficiently large to accommodate most of the HCN channels into OMDs. In this case, further increasing levels of accessible cholesterols causes a shift in the G-V relationship without affecting its slope, likely by altering the coupling between the voltage sensor and the channel pore[64,65]. Further research is needed to fully understand how different cholesterol pools influences the electromechanical coupling of HCN channels[66].

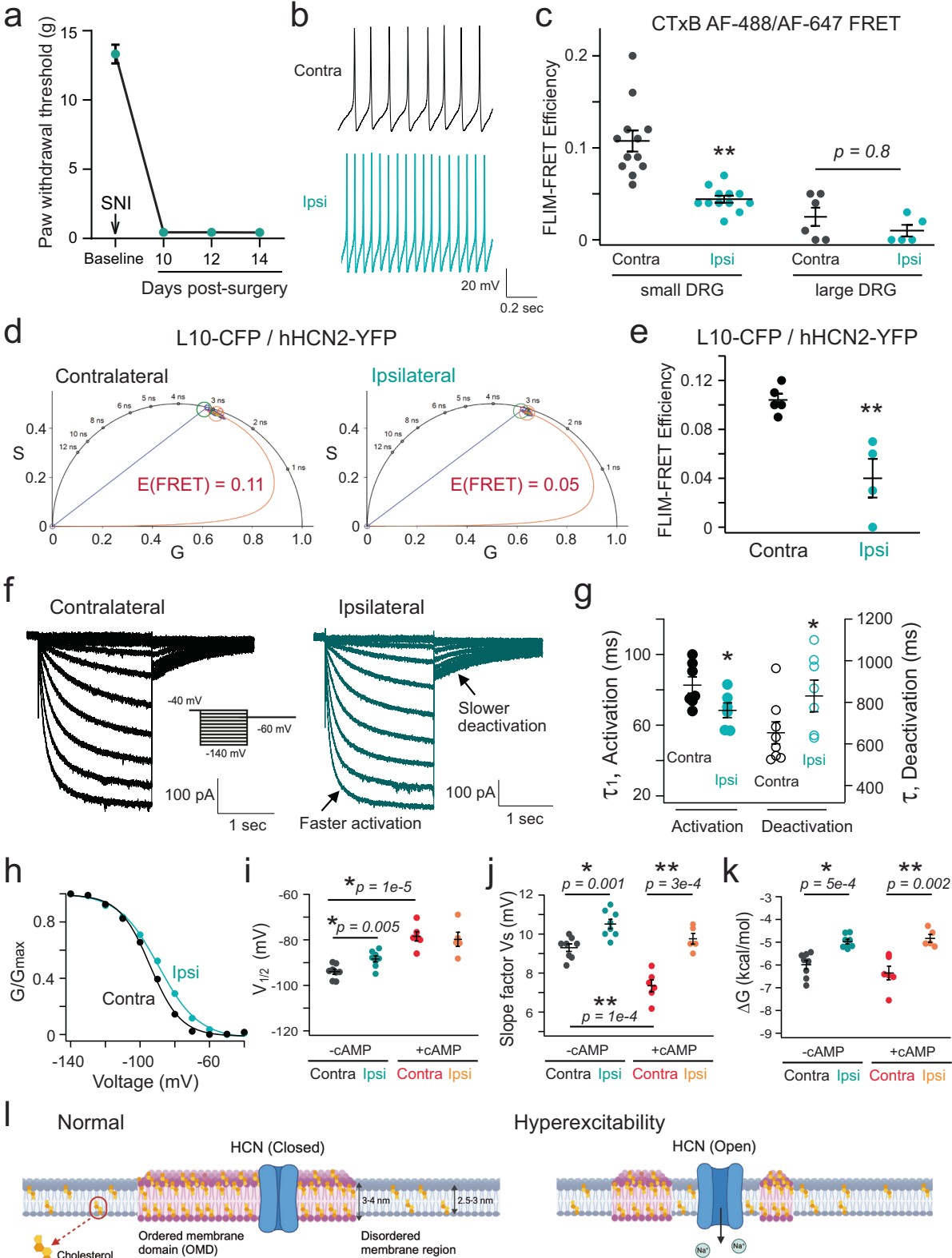

## Discussion

Neuropathic pain[67,68], a prevalent and debilitating condition, arises from nerve injury, infections, diabetes, and frequently, "channelopathies." Despite its widespread occurrence, effective treatments for neuropathic pain are currently lacking. To manage severe neuropathic pain, opioids are commonly prescribed; however, their use carries the risk of addiction or overdose[67,69]. Recently, pacemaker HCN channels have emerged as promising therapeutic targets for addressing

neuropathic pain[23,70,71]. This current study not only enhances our understanding of HCN channels, but also provides insights into potential strategies to combat neuropathic pain. Additionally, we investigated a previously overlooked aspect for the pain research: how membrane compartmentalization, particularly lipid nanodomains like OMDs, influences ion channels[3]. Our research pinpointed a modulation of pacemaker HCN channels by OMDs, which is dependent on the altered rearrangement of voltage sensors in distinct lipid domains[30]

**Fig. 7 | Assessing the OMD size and HCN2 channel localization in DRG neurons using a spared nerve injury (SNI) model of neuropathic pain. a** Paw withdrawal thresholds of rats ($n = 6$) before and after (day 0 onwards) SNI surgery, mean ± s.e.m. **b** Action potentials of neurons from contralateral and ipsilateral sides. Same cells are shown in panel (**f**) and (**h**). **c** Summary data of FRET efficiencies between CTxB probes in contralateral versus ipsilateral DRG neurons, mean ± s.e.m., $n = 6$ for contralateral and $n = 5$ for ipsilateral side large DRG neurons, $n = 12$ for small DRG neurons, **$p = $2e-5, two-sided student's $t$-test. **d** Representative phasor plots showing FRET between L10-CFP and hHCN2-YFP in contralateral versus ipsilateral DRG neurons. **e** Summary of the L10-CFP/hHCN2-YFP FLIM-FRET, mean ± s.e.m., $n = 5$ for contralateral and $n = 4$ for ipsilateral side neurons, **$p = 0.004$ using two-sided student's $t$-test. **f** HCN currents elicited by hyperpolarizing voltage pulses as in Fig. 5a, comparing small DRG neurons of contralateral and ipsilateral sides. The activation time constants are $\tau_1 = 95$ ms, $\tau_2 = 453$ ms for the contralateral and

$\tau_1 = 68$ ms, $\tau_2 = 353$ ms for the ipsilateral neuron at $-140$ mV. The deactivation $\tau$ is 763 ms for the contralateral and 867 ms for the ipsilateral neuron. **g** Summary of time constants for the activation and deactivation of HCN currents, as in panel (**f**), mean ± s.e.m. Activation: $n = 7$ cells for contralateral side and $n = 6$ for ipsilateral side, *$p = 0.04$; deactivation: $n = 8$ cells for contralateral side and n = 7 for ipsilateral side, *$p = 0.04$, two-sided student's $t$-test. **h** Representative G-V relationships of HCN channel activation of small DRG neurons. **i–k** Summary of the effects on HCN gating parameters: $V_{1/2}$, slope factor, and free energy for channel activation, mean ± s.e.m., $n = 8$ cells for both sides without added cAMP, and $n = 6$ for the contralateral side, and n = 5 for the ipsilateral side with 0.5 mM cAMP, two-sided student's $t$-test. **l** Cartoons depicting HCN channel localization in OMDs and their role in modulating DRG neuron excitability in neuropathic pain. Created in BioRender. Handlin, L. (2024) BioRender.com/q00t481.

(Fig. 6h). This finding highlights OMDs and HCN channel voltage sensors as promising therapeutic targets for achieving pain relief.

During evolution, some cells, particularly neurons, developed the ability to carry out rapid, repetitive action potential firing. Presumably, this ability required a host of biochemical and biophysical adaptations. Our research suggests that electrically active cell membranes may require larger OMDs as a protective and adaptive mechanism to maintain a stable membrane structure that supports or accommodates repetitive action potential firing. In this study, we observed that small nociceptive DRG neurons possess larger OMDs than large DRG neurons. This finding is consistent with the fact that a low percentage (~15%) of medium and large DRG neurons display spontaneous firing under our experimental condition in the presence of NGF in the culture medium. On the other hand, a significant percentage (~40%) of small DRG neurons exhibit spontaneous firing. This correlation supports the idea that OMDs play a role in adapting to the electrical demands of the cell. Our hypothesis gains further support from the data collected from five other types of cells tested. For instance, cardiac pacemaker cells, which fire rhythmic action potentials, display the largest OMD size among the non-neuronal cell types examined in this study. Future research is needed to more thoroughly examine the concept that the lipid composition and compartmentalization of the cell membrane adapt in response to distinct electrical requirements of the cell.

Our FLIM-FRET methodologies, specifically the use of the CTxB probes for quantifying the properties of OMDs of living cells, can be interpreted within the context of thermodynamic principles governing the phase separation in heterogeneous lipid membranes[43,72,73]. This biophysical concept involves a two-dimensional square-lattice Ising model to understand the critical liquid-ordered to liquid-disordered transition within the membrane[74]. Critical fluctuations in lipid membranes, measured using the fluorescence autocorrelation function, dramatically change near the miscibility critical point[43]. The correlation length of the critical fluctuations can report the size and lifetime of phase-separated domains[43,75]. In this framework, a heightened FRET signal, measured between the CTxB-based FRET pairs, indicates an increased correlation length and enlarged OMDs. A greater correlation length amplifies the likelihood of multiple GM1 gangliosides in the OMDs to facilitate CTxB binding. Additionally, it elevates the probability of CTxB molecules being in proximity to one another, thereby facilitating FRET.

Our study establishes connections between the pacemaker HCN channel, membrane compartmentalization, and neuropathic pain. We speculate that neurotrophic factors and cytokines released from injured peripheral nerves or during paclitaxel treatment could lead to altered cell signaling, compromised membrane integrity, and decreased OMD size. This, in turn, affects the trafficking and localization as well as the function of ion channels including voltage-gated sodium channels, as previously observed[59,76]. Here, our findings demonstrate that disrupting OMDs makes HCN channels more conductive and sensitive to voltage. In a separate study, we showed that

the integrity of the OMD plays a crucial role in the rearrangement of voltage sensors[30]. We believe that the movement of the unusually long S4-helix of HCN channels is highly sensitive to the "hydrophobic (mis) match" with the lipid bilayer's hydrophobic region. Considering that this hydrophobic mismatch often plays a critical role in influencing the tilt angle of transmembrane helices[77,78], we hypothesize that the loss of the localization of HCN channels in OMDs increases the tilt angle of the S4 helix. This phenomenon can be attributed to the difference in lipid bilayer thickness comparing OMDs with disordered membrane regions[5]. Consequently, this increase in the tilting motion of the S4 helix facilitates pore domain activation through its interaction with the channel pore domain[79,80]. Interestingly, disrupting OMDs leads to reduced charge movement across the electric field of HCN channels, consistent with an alteration in voltage-sensor movement of HCN channels, or alternatively, suggesting a more focused electric field of the membrane. Similar changes of the gating charge movement could happen for other types of voltage-gated ion channels when OMDs are disrupted[15]. The modulation of ion channels by OMDs and cholesterol stands as an important feature in neuronal regulation. Notably, this modulatory influence on GABA$_A$ receptors[81] and TREK potassium channels[6,82] contributes to molecular mechanisms underlying general anesthesia. Our observations support the notion that changes in OMDs have significant implications for ion channel function and contribute to the development of chronic neuropathic pain.

Eukaryotic cells evolved cholesterol to stabilize their plasma membranes. Cholesterol, embedded within the hydrophobic region of the membrane, presents challenges in studying its precise levels, dynamics, and regulatory effects on other proteins. This study highlights the essential role of membrane cholesterol in reducing neuronal hyperexcitability and modulating pain, through the inhibition of various voltage-gated ion channels, particularly HCN channels and voltage-gated sodium channels[15,76,83]. Our findings suggest that membrane cholesterol regulates HCN channels via a dual mechanism: by increasing the size of OMDs and through potential direct binding to the channels[63]. These mechanisms may differentially influence the activation curve of HCN channels, with one primarily affecting the slope and the other causing a shift toward more hyperpolarized voltages. This dual regulatory effect could also account for the variability in HCN channel gating kinetics and activation curves observed under different conditions, such as when comparing CIPN and SNI pain models. Future research will focus on distinguishing the direct effects of accessible cholesterols, abundant in the inner leaflet of the membrane, from those of OMDs. Considering cholesterol's inherent role in OMD structure, developing methods to manipulate OMD size without altering cholesterol content will be important for this direction.

## Methods
### Ethical statement
Experiments were done in accordance with protocols approved by the Institutional Biosafety Committee (IBC) of Saint Louis University. All

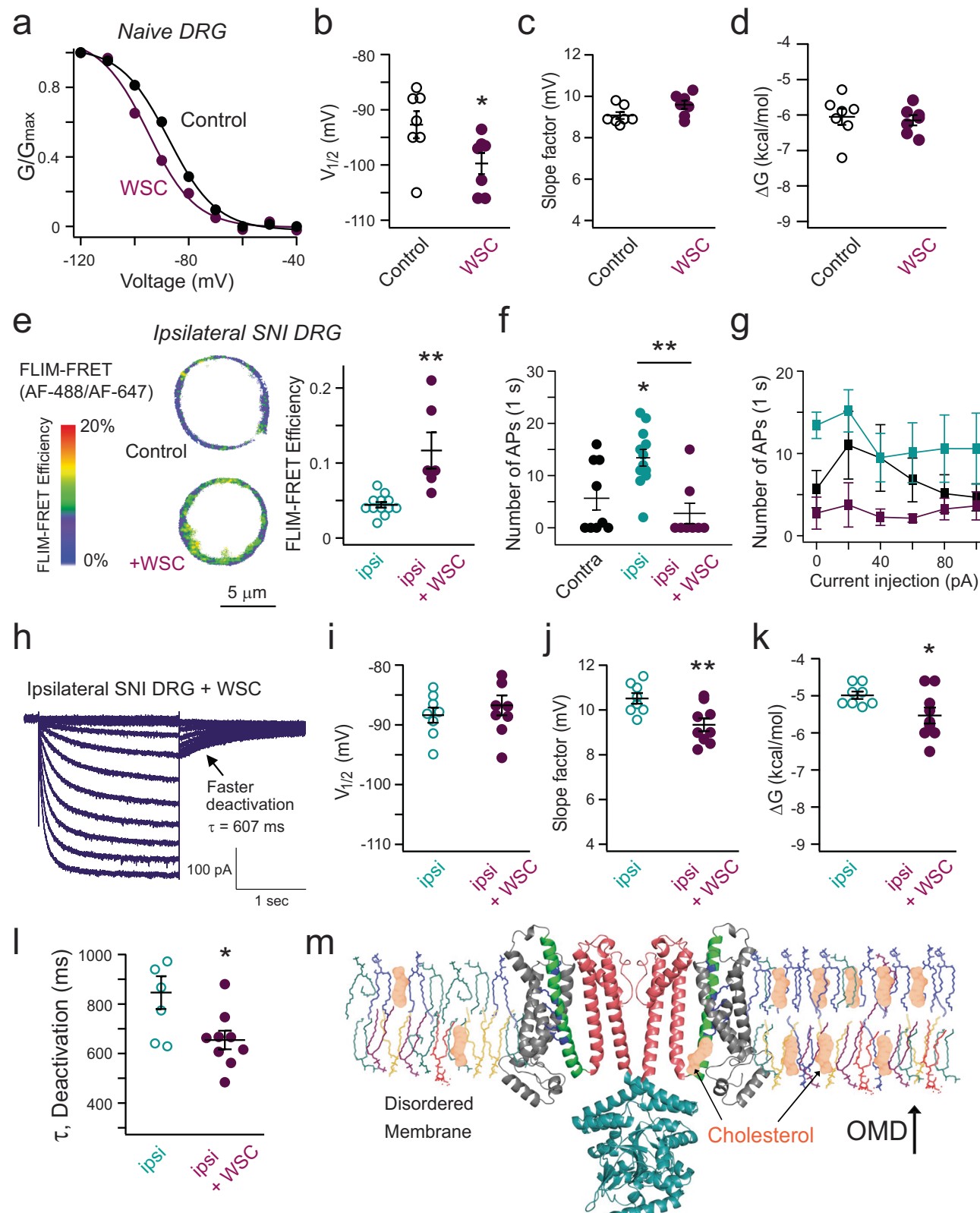

use of rats was conducted in accordance with the National Institutes of Health guidelines, and the study was conducted in strict accordance with recommendations in the guidelines approved by the Institutional Animal Care and Use Committee (IACUC) of Saint Louis University. Regarding mouse studies, all experiments involving the isolation of primary cells from mice were conducted in accordance with the guidelines approved by the IACUC of Saint Louis University, adhering to the practices outlined in the Guide for the Care and Use of Laboratory Animals.

**Molecular biology, cell culture and transfection**
Wild-type human HCN2 construct was synthesized by VectorBuilder (Chicago, IL). This construct was amplified using a mammalian gene expression vector containing a carboxyl-terminal YFP and a CMV

**Fig. 8 | Cholesterol supplementation has dual effects on HCN channel gating and attenuates neuronal excitability in SNI pain models. a** Representative G-V relationship of HCN channel activation of naive small DRG neurons with and without WSC treatment. **b–d** Summary of the effects of WSC treatment on HCN gating parameters of naive DRG neurons: $V_{1/2}$, slope factor, and free energy for channel activation, as in panel (**a**), mean ± s.e.m., $n = 7$ cells, $*p = 0.04$ using two-sided student's $t$-test. **e** WSC treatment increased the CTxB-based FLIM-FRET efficiency in ipsilateral small DRG neurons, mean ± s.e.m., $n = 12$ cells without WSC and $n = 6$ with WSC, $**p = 7e-4$ using two-sided student's $t$-test. **f** Summary of the effects of WSC on action potential firing in ipsilateral small DRG neurons. Data shown are mean ± s.e.m., $n = 9$ for contralateral side neurons, $n = 12$ for ipsilateral side neurons ($*p = 0.02$), $n = 8$ for ipsilateral neurons with WSC, $**p = 0.002$ comparing with and without WSC for ipsilateral side neurons, one-way ANOVA (no adjustment). **g** Summary of the number of action potentials of small DRG neurons versus the amplitude of current injection (same sample size as in panel (**f**), 0 pA current

injection), mean ± s.e.m. With the +20 pA current injection, $*p = 0.04$ comparing with and without WSC for ipsilateral neurons. No statistical significance was detected in other amounts of current injection. **h** HCN currents elicited by hyperpolarizing voltages in WSC-treated ipsilateral small DRG neurons. **i–k** Summary of gating parameters of HCN currents with ($n = 9$ cells) and without ($n = 8$ cells) WSC treatment for ipsilateral small neurons in the SNI model: $V_{1/2}$ ($p = 0.45$), slope factor ($**p = 0.007$), and free energy ($*p = 0.04$) for channel activation, mean ± s.e.m., two-sided student's $t$-test. **l** Summary data of HCN deactivation $\tau$, with ($n = 9$) and without ($n = 7$) WSC treatment for ipsilateral small DRG neurons, mean ± s.e.m., $*p = 0.02$ using two-sided student's $t$-test. **m** Schematic illustrating cholesterol's dual effects on HCN channel gating: via OMD expansion or direct binding. HCN3-based structure (PDB: 8INZ) and membrane lipids are shown in ordered (right) vs. disordered (left) states. Created in BioRender. Dai, G. (2024) BioRender.com/f24v785.

promoter. The human HCN2-L323TAG/T240TAA (two Opal TGA stop codon used as the functioning stop codon, ID: VB231116-1293ket) was also made by VectorBuilder (Chicago, IL). The L10 and S15 fluorescent probes are gifts from Prof. Bertil Hille (University of Washington, Seattle, WA). DNA concentration was measured using a Nanodrop OneC spectrophotometer (Life Technologies, Grand Island, NY).

The tsA-201 cells (a variant of human embryonic kidney HEK cells, catalog #: 96121229) were obtained from Sigma-Aldrich (St. Louis, MO). Cells were cultured in Dulbecco's modified Eagle's medium (DMEM; Gibco) supplemented with 10% fetal bovine serum (FBS; Gibco) and 1% penicillin/streptomycin (Gibco) in a humidified incubator at 37 °C with 5% $CO_2$ in tissue culture dishes (CellTreat, Pepperell, MA). Transfection was performed on 70%–90% confluent tsA-201 cells using the lipofectamine 3000 Kit (Invitrogen, Carlsbad, CA, #L30000080), according to the manufacturer's protocol[47,84]. We employed lipofectamine 3000 and electroporation transfection techniques, to introduce L10 and S15 probes into DRG neurons. We found that the lipofectamine method was more favorable for transfecting small DRG neurons, whereas electroporation exhibited higher efficiency with larger DRG neurons. For primary mouse hepatocytes, we used lipofectamine 3000 for the transient transfection of L10 probes. Additionally, we found some large-sized primary cultured cells are resistant to lipofectamine-mediated transfection, e.g., fibroblasts were not able to be transfected using the lipofectamine method. The hypotonic solution was made by adjusting to 206 mOsm osmolarity through the addition of sucrose to a solution made of 80 mM NaCl, 3 mM KCl, 1 mM $CaCl_2$, 1 mM $MgCl_2$, 5 mM glucose, and 10 mM HEPES, pH 7.4.

The cholera toxin subunit B-Alexa Fluor™ 488 and 647 conjugates were obtained from Thermo Fisher Scientific and applied to cultured cells at a concentration of 20 nM for a duration of ~10 min. The binding affinity of CTxB to GM1-enriched membrane falls within the picomolar range[35]. β-cyclodextrin and water-soluble cholesterol were purchased from Sigma-Aldrich (St. Louis, MO).

The approach of dual stop-codon suppression was developed based on prior research[30,61]. The tsA-201 cells were co-transfected with two plasmids: pAS_4xU6-PyIT M15(UUA) FLAG-Mma PyIRS (Mma PyIT/RS) and pAS_4xhybPyIT A41AA C55A FLAG-G1 PyIRS Y125A (G1PyIT/RS) tRNA/aminoacyl-tRNA synthetase pairs from Addgene (Watertown, MA, #154774 and #154773 respectively), along with a third plasmid containing the hHCN2 channel construct with dual stop-codon mutations for selective incorporation of trans-cyclooct-2-ene-L-lysine (TCO*K) and N-propargyl-L-lysine (ProK) (SiChem, Bremen, Germany). At the TCO*K site, IEDDA reaction was performed by applying 1–2 μM tetrazine dye (Click Chemistry Tools, Scottsdale, AZ) to the culture medium and incubating it with the cells at 37 °C for 30 min. At the ProK site, the copper-catalyzed azide-alkyne cycloaddition reaction was performed using 50 μM $CuSO_4$, 250 μM THPTA (Tris(benzyltriazolylmethyl)amine), and 2.5 mM ascorbic acid. Cells were

incubated with 5 μM picolyl azide dye (Click Chemistry Tools, Scottsdale, AZ) for 10 min at room temperature.

## Fluorescence Lifetime Measurement and FLIM-FRET

A Q2 laser scanning confocal system, equipped with a FastFLIM data acquisition module from ISS, Inc. (Champaign, IL), and two hybrid PMT detectors (Model R10467-40, Hamamatsu USA, Bridgewater, NJ), was used for digital frequency-domain fluorescence lifetime imaging[30,85]. This confocal FLIM system was connected to an Olympus IX73 inverted microscope (Olympus America, Waltham, MA). The fluorescence lifetime measurements provided both frequency-domain (instantaneous distortion-free phasor plotting) and time-domain decay information. Confocal imaging was made possible using dichroic cubes, one with a 50/50 beam splitter and the other equipped with various long-pass filters. A supercontinuum laser (YSL Photonics, Wuhan, China) was used for fluorescence excitation, with a wavelength range of 410 nm to 900 nm, a repetition rate of 20 MHz, and a pulse duration of 6 ps. Specifically, the excitation wavelengths were 445 nm for CFP, 488 nm for AF-488 and YFP, 561 nm for AF-555. The intensity of laser excitation was kept at the same level when imaging different cell samples. To detect the emission from the CFP/YFP FRET pair, an emission filter cube comprising a 475/28 nm filter for CFP, a 495 nm long-pass dichroic mirror, and a 542/27 nm filter for YFP was used. For the AF-488/AF-555 FRET pair, an emission filter cube consisting of a 525/40 nm filter for AF-488, a 552 nm long-pass dichroic mirror, and a 593/40 nm filter for AF-555. For phasor FLIM-FRET, only the donor lifetime was measured either in the absence or in the presence of the FRET acceptor. In the FRET experiments related to L10 and S15, the intensity-based ratio of FRET donors to acceptors is randomized, and no significant differences were observed for this ratio under various conditions.

Under each experimental condition, confocal images with a frame size of 256 × 256 pixels were acquired. A motorized variable pinhole with a size of 100 μm and a pixel dwell time ranging from 0.1 to 0.4 ms were used during image acquisition. For image processing, display, and acquisition, the VistaVision software (ISS, Inc., Champaign, IL) was used. This software allows for the customization of parameters such as pixel dwell time, image size, and resolution. The FLIM analysis used a fitting algorithm, phasor plotting and analysis. To enhance the display, isolation, and analysis of membrane-localized lifetime species, a median/gaussian smoothing filter and an intensity-threshold filter were applied. The phasor FLIM approach offers a notable advantage in effectively distinguishing various lifetime species within the phasor plot. It allows for a separation of cytosolic fluorescence and background fluorescence, from the lifetime species originating from the membrane-localized fluorescence signals. Specifically, lifetime species from endocytosed CTxB that are localized inside of the cell can be separated from those from the membrane-localized CTxB (Supplementary Fig. 1b). The lifetime of the membrane-localized FRET donor

CTxB AF-488 is unaffected by treatments of β-CD or detergents (Supplementary Fig. 1c–e). Overall, the fluorescence intensity of endocytosed CTxB is minor, and membrane-localized fluorescence is predominant. In addition, the fluorescence intensity of CTxB AF-488 labeling showed minimal to no correlation with the measured FLIM-FRET efficiencies.

The VistaVision software was used to determine the phase delays (φ) and modulation ratios (m) of the fluorescence signal in response to an oscillatory stimulus with frequency ω. These values were derived by conducting sine and cosine Fourier transforms of the phase histogram, considering the instrument response function (IRF) calibrated with different fluorophores with known lifetimes in water. The choice of calibration fluorophore (Atto 425, rhodamine 110, or rhodamine B) depended on the emission spectrum of the fluorophore of interest, with respective lifetimes of 3.6 ns, 4 ns, and 1.68 ns.

The FLIM-FRET analysis was performed using the FRET trajectory function available in the VistaVision software[30,39]. To ensure accurate results, three crucial parameters were adjusted to optimize the fitting trajectory, encompassing both the unquenched donor and the donor species affected by FRET. These parameters included: the background contribution in the donor sample, the unquenched donor contribution in the FRET sample, and the background contribution in the FRET sample. To maintain accuracy, the background levels in both the donor sample and the FRET samples were set below 4%. This value corresponds to the very low emission intensity typically observed in unlabeled or untransfected cells.

## Patch-clamp electrophysiology
The whole-cell patch-clamp recordings were performed using an EPC10 patch-clamp amplifier and PATCHMASTER (HEKA) software, with a sampling rate of 5 or 10 kHz. Borosilicate patch electrodes were created using a P1000 micropipette puller from Sutter Instrument (Sutter, Novato, CA), resulting in an initial pipette resistance of ~3–5 MΩ. A Sutter MP-225A motorized micromanipulator was used for patch clamp (Sutter, Novato, CA). The recordings were conducted at a temperature range of 22–24 °C.

To record HCN currents from DRG neurons, the internal pipette solution was prepared with the following composition (in mM): 10 NaCl, 137 KCl, 1 MgCl₂, 10 HEPES, 1 EGTA, and pH 7.3 adjusted with KOH. In addition, 0.5 mM cAMP was added to the internal solution to facilitate the endogenous HCN currents. The external solution used was prepared with the following composition (in mM): 154 NaCl, 5.6 KCl, 1 MgCl₂, 1 CaCl₂, 8 HEPES, 10 D-Glucose, and pH 7.4 adjusted with NaOH. Series resistance was typically lower than 10 MΩ and series resistance compensation was set at 40–70%. For the action potential measurements, current clamp mode of PATCHMASTER was employed instead of the whole-cell voltage-clamp configuration. In the current clamp mode, the resting membrane potential ($-57 \pm 1$ mV, $n = 6$ cells) was established through the application of zero current injection. The input resistance of DRG neurons, obtained by applying small negative current injections, was found much smaller than the seal resistance that is typically higher than one giga-ohm. Subsequent current injections for evoking action potentials were administered as absolute values. Neurons frequently exhibit adaptability to injected currents, contributing to the observed variability in the number of action potential firings during the test period. The adaptation to injected currents leads to a reduction in action potential firing rates as the injected current increases.

The conductance-voltage (G-V) relationships of endogenous HCN currents were measured from the instantaneous tail currents at −60 mV following voltage pulses from −40 mV to between 0 and −140 mV. The leak tail currents following pulses to −60 mV were subtracted, and the currents were normalized to the maximum tail current amplitude, reporting the relative channel conductance ($G/G_{max}$). The relative conductance was plotted as a function of the voltage of the main pulse and fitted with the Boltzmann equation:

$$G/G_{max} = 1/\left(1 + \exp[(V - V_{1/2})/V_S]\right) \quad (1)$$

where V is the membrane potential, $V_{1/2}$ is the potential for half-maximal activation, and Vs is the slope factor. $Vs = RT/zeF$, where R is the gas constant, T is temperature in Kelvin (297 K), F is the Faraday constant, e is the value of elementary charge, z is the estimated number of elementary gating charge per channel for channel activation. The difference in Gibbs free energy between the channel closed state and open state at 0 mV (or absence of an electric field) was calculated according to: $\Delta G = RTV_{1/2}/Vs$[86]. In this case, the less negative the value of $\Delta G$, the more likely the channel will open at 0 mV. Therefore, $\Delta G$, which is dependent on the gating charge z, can estimate the relative amount of free energy needed for channel activation. Although the $\Delta G$ and the z values of native HCN channels derived from ionic current recordings in primary DRG neurons are evidently underestimated, this paper emphasizes not the quantification of absolute gating charge movement or $\Delta G$, but rather the examination of directional changes in relative gating charge during channel activation under various conditions. We also noticed that the longer hyperpolarizing voltage protocol is only slightly better than the short protocol in estimating the gating charge. In addition, our measured apparent gating charge reflects the averaging effect of a heterogeneous population of HCN channel subtypes present in small nociceptor DRG neurons[18,87,88].

The cAMP-induced decrease in the slope factor of the G-V relationships is subtle but noticeable from native HCN channels of DRG neurons. This finding aligns with previous research showing that cAMP similarly decreased the slope factor in mouse HCN2[89]. However, it contrasts with a previous study suggesting that cAMP binding decreases the gating charge[90].

## Rat studies and primary DRG neurons
Adult Sprague-Dawley rats (Envigo, Placentia, CA), pathogen-free females (100 g) and males (250 g), aged 5–7 weeks, were housed under controlled conditions: a 12-h light/dark cycle (lights on at 07:00), temperature at $23 \pm 3$ °C, and humidity between 40 and 60%. Standard rodent chow and water were available ad libitum. All behavioral experiments were performed by experimenters who were blinded to treatment groups.

Dorsal root ganglia (DRG) were dissected from the female and male Sprague-Dawley rats[91–93]. In brief, removing dorsal skin and muscle and cutting the vertebral bone processes parallel to the dissection stage exposed DRGs. DRGs were then collected, trimmed at their roots, and digested in 3 mL bicarbonate free, serum free, sterile DMEM (Catalog # 11965, Thermo Fisher Scientific, Waltham, MA) solution containing neutral protease (3.125 mg/mL, catalog # LS02104, Worthington, Lakewood, NJ) and collagenase Type I (5 mg/mL, catalog # LS004194, Worthington, Lakewood, NJ) and incubated for 45 min at 37 °C under gentile agitation. Dissociated DRG neurons (~1.5 × 10⁶) were then gently centrifuged to collect cells and washed with DRG media DMEM containing 1% penicillin/streptomycin sulfate from 10,000 μg/mL stock, 30 ng/mL nerve growth factor, and 10% fetal bovine serum. Cells were plated onto poly-D-lysine and laminin-coated glass 12-mm coverslips.

## Spared nerve injury (SNI) model of neuropathic pain
Under isoflurane anesthesia (5% induction, 2.0% maintenance in 2 L/min O₂), skin on the lateral surface of the left hind thigh was incised. The biceps femoris muscle was bluntly dissected to expose the three terminal branches of the sciatic nerve[62]. Briefly, the common peroneal and tibial branches were tightly ligated with 4-0 silk suture and axotomized 2.0 mm distal to the ligation. Closure of the incision was made in two layers. The muscle was sutured once with 5-0 absorbable suture;

skin was auto-clipped. Animals were allowed to recover for 7 days before any testing. Both male and female animals were used.

## Mouse studies

The mice used in the study were the wild-type C57BL/6J strain and were bred from an in-house colony. The mice were housed in a specific pathogen-free animal facility at Saint Louis University, where they were subjected to a 12-h light/dark cycle, 22–24 °C and 40–60% humidity in ventilated cages with. Ad libitum access to drinking water was provided by automatic cage waterers. Standard rodent chow was also provided ad libitum. All mice were group housed, up to five mice per cage. Mice used for cell isolation came from our in-house breeding colony, of the parental C57BL/6J strain. Both male and female mice were used for cell isolations and cells were isolated from mice 8–20 weeks of age.

## Isolation of mouse pacemaker cells

The isolation and culturing of mouse pacemaker cells was refined from the procedures outlined in a previous paper[94]. In summary, wild-type C57BL/6J mice were anesthetized by isoflurane, thoracotomy performed, and the beating hearts surgically harvested. The heart explant was rapidly immersed in warm Tyrode's solution (pH 7.4) with the following composition (in mM): 140 NaCl, 5.4 KCl, 1 $MgCl_2$, 1.8 $CaCl_2$, 5 HEPES, and 5.5 glucose. The sinoatrial node (SAN) was identified under a dissection microscope and carefully excised from the superior vena cava, sulcus terminalis, sulcus coronarius, and inferior vena cava. The SAN was then transferred to a 2-mL reaction tube containing 675 µl of pre-warmed Tyrode's low-$Ca^{2+}$ solution (pH 6.9) with the following composition (in mM): 140 NaCl, 5.4 KCl, 0.5 $MgCl_2$, 0.2 $CaCl_2$, 5 HEPES, 5.5 glucose, 1.2 $KH_2PO_4$, and 50 taurine. After a 5-min incubation, enzymatic digestion of the SAN was initiated by adding 10 µl of BSA (1 mg/mL), 40 µl of elastase (18.87 U), 125 µl of protease (1.79 U), and 150 µl of collagenase B (0.54 U) to the 2-mL reaction tube. The digestion process occurred for 30 min in a 37 °C water bath, with additional mechanical dissociation performed every 10 min. The reaction was stopped by centrifugation at 200 x $g$ for 2 min at 4 °C, and the supernatant was discarded. The SAN was then washed three times with 1 mL of ice-cold, calcium-free Kraft-Brühe (KB) solution (pH 7.4) with the following composition (in mM): 80 L-glutamic acid, 25 KCl, 3 $MgCl^{2+}$, 10 $KH_2PO_4$, 20 taurine, 10 HEPES, 10 glucose, and 0.5 EGTA.

After the washing steps, the SAN tissue was allowed to rest in 350 µl of cold KB solution at 4 °C for up to 3 h. Following the recovery period, the SAN was incubated in a water bath at 37 °C for 10 min. The SAN cells were mechanically dissociated using a 1000 µl pipette tip and further dissociated with a 200 µl pipette tip. 50 µl of dissociated cells were plated on poly-L-lysine-coated 25 mm coverslips within a 100 mm × 15 mm petri dish containing moistened paper towels to maintain humidity. The cells were allowed to attach to the coverslip. Before proceeding with experimentation, the cells were returned to physiological conditions, and their automaticity was restored by gradually reintroducing $Ca^{2+}$. This was achieved by adding Tyrode's solution (pH 7.4) in small increments (10, 22, 50, and 172 µl) every 3 min.

## Primary hepatocyte isolation and culture

Primary murine hepatocytes were isolated following a two-step perfusion method[95]. Mice were euthanized using deep isoflurane anesthesia followed by cervical dislocation. The abdominal cavity was promptly exposed, and a 22-gauge intravenous catheter was cannulated into the portal vein. The liver was initially perfused with Hank's buffered salt solution (HBSS, Gibco) supplemented with 0.5 mM EGTA at a flow rate of 5 mL/min to clear blood from the hepatic vasculature. Simultaneously, an incision was made in the inferior vena cava to drain the perfusate and prevent over-pressurization. After 4-5 min of HBSS perfusion, the pump tubing was switched to perfuse serum-free

Dulbecco's modified eagle's medium (DMEM; Sigma) containing 1 mg/mL type IV collagenase (Sigma). The collagenase perfusion lasted for another 4-5 min, allowing for liver digestion. Both the HBSS and collagenase solutions were warmed to 37 °C before and during perfusion. Following the removal of the gallbladder, the digested liver was transferred to a 10-cm petri dish containing 10 mL of collagenase-DMEM solution, and the capsule layer was manually removed using forceps. The digested liver lobes were then dispersed by pipetting up and down with a 25 mL serological pipette. The cell suspension was supplemented with 15 mL of ice-cold 10% FBS-DMEM, passed through a 100 µm cell strainer, and kept on ice. Cells were pelleted at $50 \times g$ for 2 min, washed with fresh 10% FBS-DMEM, recentrifuged, and suspended again in fresh 10% FBS-DMEM. Cell counting was performed using a hemocytometer, and the cells were plated on type-I collagen-coated glass-bottomed dishes at a concentration of 200,000 cells/mL in 10% FBS-DMEM supplemented with penicillin/streptomycin and amphotericin B. The cells were allowed to adhere to the dishes for 3–4 h before being cultured overnight in basal DMEM (glucose, serum, and phenol-red free; Sigma) supplemented with either 300 µM BSA-palmitate, 200 µM BSA-palmitate plus 100 µM BSA-oleate, or equimolar fatty-acid free BSA (BSA-conjugated fatty acids from Cayman Chemical; 6:1 fatty acid to BSA ratio, catalog # 29557 and 29558) in a 5% $CO_2$ incubator at 37 °C. Imaging of the cells was performed the following day.

## Dermal fibroblast isolation and culture

Adult murine dermal fibroblasts were isolated from the ear lobes of mice using a humane method involving deep isoflurane anesthesia followed by cervical dislocation. The ear tissue was minced with a razor blade and placed into wells of a 6-well plate. It was then digested overnight in a 5% $CO_2$ incubator at 37 °C using 500 units of collagenase in 20% FBS-DMEM supplemented with penicillin/streptomycin. The following day, the cells were dispersed by pipetting up and down with a 5 mL pipette. The cell suspension was transferred into a 15 mL conical tube, centrifuged at $300 \times g$ for 5 min, and the cell pellet was resuspended in 5 mL of fresh 20% FBS-DMEM. The cells were then plated onto a 6-cm dish. Once the fibroblasts reached confluency, they were passaged by trypsinization. For imaging purposes, cells at passages 2–3 were used, and they were seeded onto glass-bottomed dishes at a concentration of 100,000 cells/mL.

## H9c2 cell culture

H9c2 embryonic rat cardiomyocyte cells were obtained from the American Type Culture Collection (ATCC, Manassas, Virginia, catalog #: CRL-1446) and cultured in 10% FBS-DMEM until ~90% confluent before passaging by trypsinization. Cells were used at passages 3–10. Cells were plated on glass-bottomed dishes at 100,000 cells/mL for imaging.

## Xenopus oocyte

Isolated oocytes were purchased from EcoCyte (Austin, TX) and cultured in a saline-like solution ND96 (96 mM NaCl) supplemented with antibiotics (100 U/mL penicillin/100 µg/mL streptomycin).

## Statistics and reproducibility

The data parameters were presented as the mean ± s.e.m. of n independent cells or patches. Statistical significance was determined using the unpaired or paired two-sided $t$-test for comparing two groups, or a one-way ANOVA followed by Tukey's post hoc test for comparisons with more than two independent groups, denoted as *$p < 0.05$, **$p < 0.01$ respectively.

## Reporting summary

Further information on research design is available in the Nature Portfolio Reporting Summary linked to this article.

## Data availability

Source data supporting Figs. 1–8 and Supplementary Figs. are provided as a Source Data file (also in FigShare [https://doi.org/10.6084/m9.figshare.27183531.v1]), and additional data are in the Supplementary Information. Please refer to [https://doi.org/10.2210/pdb8INZ/pdb] for the PDB accession code 8INZ used in this paper. Source data are provided with this paper.

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

## Acknowledgements

We thank Donald W. Hilgemann (University of Texas Southwestern Medical Center, Dallas) for advice and editing the manuscript, Bertil Hille (University of Washington, Seattle) for sharing DNA constructs, Joel C. Eissenberg for editing the manuscript, Christine Kim and Samantha Deavila for technical support. This research is supported by startup funds from the Edward A. Doisy Department of Biochemistry and Molecular Biology, Saint Louis University School of Medicine (to G.D. and K.S.M), and National Institute of General Medical Sciences R35GM154778 grant (to G.D.). A.M. is supported by startup funds from the Department of Pharmacology and Physiology, Saint Louis University, SLU Institute for Drug & Biotherapeutic Innovation seed grant, National Institute of Neurological Disorders and Stroke R01NS119263, R01NS119263-04S1 grants, and Department of Defense CPMRP #CP220080P1. Cartoons in figures were created using BioRender (CC-BY publication license obtained).

## Author contributions

G.D. conceptualized the research. G.D., A.M., and K.S.M. designed experiments. L.J.H. performed electrophysiology experiments, G.D. and N.L.M. performed fluorescence imaging. N.L.A.D., L.S., and A.M. performed behavior experiments on rats and prepared neuronal samples. K.S.M. and G.D. prepared primary cardiac pacemaker cells. K.S.M. prepared other primary cells, including hepatocytes and fibroblasts. G.D., L.J.H., N.L.M., and N.L.A.D. analyzed data. G.D. and E.N.L. created figures. G.D. wrote the paper with inputs from other authors.

## Competing interests

The authors declare no competing interests.
