## [Transparent Peer Review file · Nature Communications]

Membrane Lipid Nanodomains Modulate HCN Pacemaker Channels in Nociceptor DRG Neurons

Corresponding Author: Dr Guacan Dai

Version 0:

Reviewer comments:

Reviewer #1

(Remarks to the Author)

Membrane lipids play a crucial role in controlling the function of ion channels, and compartmentalization of ion channels within lipid domains further regulates their properties. This is also true for HCN channels, as previously shown also by the Authors that have set up an impressive wealth of experimental approaches that address many questions, related to the compartmentalization of channel in liquid ordered domains of the membrane. In this paper they apply the same approach, based on technically demanding FRET-FLIM, to study lipid-induced effects on localization and intramolecular movements of HCN2. This channel is involved in chronic pain and it is known that it is activated in inflamed /altered DRG somatosensory neurons in which it localizes.

There are several interesting and valuable results, including the high presence of Liquid ordered domain (Lo or OMD) in cardiac pacemaker cells (where HCN4 is highly expressed) as well as in DRG of small diameter, where HCN2 is found. OMD disruption by β -CD increases firing in DRG and the Authors claim that it is correlated with some evidence of increased open probability in HCN2 channels. Also, in animal models of neuropathic pain, they find that OMD size is reduced and HCN2 is localized in liquid disordered regions of the membrane.

It is tempting to speculate, and this is indeed the main message of the paper, that alteration of OMD in pathologic neurons and re-localization of HCN2 channel are associated with increased firing activity in DRG neurons. Overall, it is a very interesting paper and I greatly enjoyed reading it. The point of view that lipid environment can modulate HCN2 channel and contribute to neuropathic pain is novel, valuable, and worth addressing. The Authors provided a clear demonstration, both in present and previous work (Handlin et al., Nat Communication 2023), that both HCN2 and HCN4 like to reside in OMDs, while HCN1 does not, and that OMDs disruption (i.e., β -CD treatment) similarly affected the voltage sensor movement of both HCN2 and 4 isoforms, while SVD movement of HCN1 is insensitive to OMDs disruption. The authors, indeed, stated that "This reduction (FRET on β -CD treated cells expressing HCN2, Fig. 5) closely mirrored the changes in FRET between corresponding sites in HCN4 channels (Handlin et al., Nat Communication 2023). In contrast, minimal FRET change was detected between comparable sites in HCN1 channels following β -CD treatment, as shown previously (Handlin et al., Nat Communication 2023).

However, I found some weaknesses in the electrophysiology results that in my opinion are overinterpreted and, as they are, do not support the main claim. The evidence for HCN2 activation, due to an increase in open probability, is not convincing and requires some explanations but in my opinion also additional experiments.

Figure 5-a-c: It is surprising that β -CD does not shift the activation curve of HCN2, while it does (about +11 mV) in HCN4 (Barbuti et al, 2006). Authors need to address this difference.

Figure 5C- the change in slope is truly minimal and can be due to many other experimental problems rather than being a real effect. For instance, as one can judge from the very few original data shown, the one in Figure 5a shows that the voltage protocol is not adequate to properly record HCN2 current as it does not allow to reach full activation state of the current at depolarized voltages. The statistics of Figure 2E shows that the difference, converted in apparent gating charge, is barely significant. It seems odd to draw all the conclusions on the increase in P_o at -60 mV from this puzzling set of results. Increasing n , the number of recordings, is definitively needed, as well as improving the protocol, as mentioned above, and most important, showing the original data of the currents. It would also help to include a table with individual $V_{1/2}$ and slope values from the Boltzmann fitting.

Figure 5e-

- The gating charge value seems low. Is this comparable with other z values reported in the literature?
- The gating charge does not change with cAMP. This is in contrast with other approaches showing that it changes, (Hummert et al, 2018 DOI: 10.1371/journal.pcbi.1006045) and should at least be commented on.
- It would be important to show raw data for all treatments, given the small effect we are dealing with. Authors should show exemplary currents recorded after the treatment with β -CD, paclitaxel, with and without cAMP.

Figure 5f

- shows that ΔG is the same without and with cAMP. How can it be? cAMP binding provides the energy to facilitate voltage gating, and ΔG should be reduced.

The Authors use the following formula to calculate $\Delta G = RT V_{1/2}/V_s$
with $V_{1/2}$, half activation voltage
 V_s , the slope factor,
RT are constant

Since $V_{1/2}$ changes, ΔG cannot be the same in both cases.

It would be also helpful to extend the analysis of ΔG of activations also to HCN1 and HCN4. Indeed, ΔG values are used in the present work to corroborate the assumption that there is an OMD-dependent movement of HCN2 VSD. Therefore, a comparison of ΔG of activations among the three HCN subtypes will strengthen the conclusions of the work. To do not burden the main figures, I suggest to add a Suppl. Figure with ΔG of activations obtained from the analysis of HCN4 and HCN1 activation curves.

Minor changes:

- Please, use isotopes instead of isoforms.
- Please, organize Suppl. Figures based on their appearance in the text.
- Suppl. Fig. 5b and c. Statistics is absent, though significance of the treatments is highlighted in the text. Please, add statistics, and, if there is not significance, modify the text accordingly. Representative firings in panel A seems to indicate a difference. In case this is not statistically significant, please, explain why.
- From the electrophysiological point of view, there is clear difference between small and large DRG, as the first express a slow and cAMP-sensitive current, while the second a fast and insensitive one. In explaining this difference, I suggest to refer to the seminal papers discriminating between small and large DRGs on the basis of Ih properties. Nonetheless, it may be worth discriminating small DRGs also for their response to capsaicin (TRPV1), or perform some staining with nociceptive-small DRG markers (Nav1.8, for instance).

Reviewer #2

(Remarks to the Author)

In this manuscript, authors investigate the role of OMDs on neuronal response and protein function. While the results are interesting, they are always in one direction, i.e., domains get smaller (by CD treatment) and resulting response is due to this change. It is necessary to show that the other way around is also possible, i.e., if you can make the OMDs bigger, the response will be the opposite. This could be done by inserting cholesterol in the membrane. Moreover, the impact of cholesterol specifically vs OMDs generally should be dissected by changing the OMD size without changing the cholesterol content. My major comments are below.

Fig 1: The observed effects can be due to the access membrane and resulting topology. To rule this out, authors should perform osmotic swelling on the cells which should not change the results if the observed effects are due to lipid domains. Some images should be show in the main Figures to judge the quality of the images, staining etc.

Figure 2: would the cell type differences be due to differential GM1 abundance? How does the brightness of the staining compare in different cell types and is there any correlation between staining and FRET efficiency?

Proper statistical tests should be applied to the data.

Figure 3: Authors should show that the opposite of this scenario is also possible: i.e., when OMDs are enhanced, the firing frequency gets smaller. Otherwise, the observed effect can be due to cholesterol but not generally OMDs.

Figure 4: Why isn't there the data from Lck construct? It is really helpful to show both of them in parallel to increase the robustness of the results.

Figure 5: How can HCN2 change its domain localization. Based on recent papers, the localization of a protein in OMDs are determined by the protein structure, post-translational modifications etc. How does a protein leave the OMDs? This requires an explanation. Again, the opposite experiment (OMD enhancement) will be useful.

Reviewer #3

(Remarks to the Author)

I co-reviewed this manuscript with one of the reviewers who provided the listed reports. This is part of the Nature Communications initiative to facilitate training in peer review and to provide appropriate recognition for Early Career Researchers who co-review manuscripts

Version 1:

Reviewer comments:

Reviewer #1

(Remarks to the Author)

The Authors have replied to all the points raised, and cleared most of them.

There is just one answer that is not completed though. When I was asking for deltaG values for HCN1 and HCN2, I meant before and after the addition of beta-CD, while the Authors in their reply have added DeltaG values in the absence of beta-CD only.

According to the hypothesis presented by the Authors after the treatment with β -CD deltaG values become more positive in HCN2 and HCN4 (because their VSD are sensitive to cholesterol) and are unchanged in HCN1, that is cholesterol insensitive. This is while it would be important to know the values.

[Also note that the values were added to the legend of Suppl Figure 10, and not of Suppl. Figure 11, as mentioned by the Authors].

Reviewer #2

(Remarks to the Author)

Authors addressed my major concerns. Some points, such as direct role of cholesterol vs OMDs, were not directly addressed, however, the authors discussed these limitations in the manuscript. Therefore, I am satisfied with the revised version.

Reviewer #3

(Remarks to the Author)

Reviewer #1 (Remarks to the Author):

Membrane lipids play a crucial role in controlling the function of ion channels, and compartmentalization of ion channels within lipid domains further regulates their properties. This is also true for HCN channels, as previously shown also by the Authors that have set up an impressive wealth of experimental approaches that address many questions, related to the compartmentalization of channel in liquid ordered domains of the membrane. In this paper they apply the same approach, based on technically demanding FRET-FLIM, to study lipid-induced effects on localization and intramolecular movements of HCN2. This channel is involved in chronic pain and it is known that it is activated in inflamed /altered DRG somatosensory neurons in which it localizes.

There are several interesting and valuable results, including the high presence of Liquid ordered domain (Lo or OMD) in cardiac pacemaker cells (where HCN4 is highly expressed) as well as in DRG of small diameter, where HCN2 is found. OMD disruption by β -CD increases firing in DRG and the Authors claim that it is correlated with some evidence of increased open probability in HCN2 channels. Also, in animal models of neuropathic pain, they find that OMD size is reduced and HCN2 is localized in liquid disordered regions of the membrane.

It is tempting to speculate, and this is indeed the main message of the paper, that alteration of OMD in pathologic neurons and re-localization of HCN2 channel are associated with increased firing activity in DRG neurons.

Overall, it is a very interesting paper and I greatly enjoyed reading it. The point of view that lipid environment can modulate HCN2 channel and contribute to neuropathic pain is novel, valuable, and worth addressing. The Authors provided a clear demonstration, both in present and previous work (Handlin et al., Nat Communication 2023), that both HCN2 and HCN4 like to reside in OMDs, while HCN1 does not, and that OMDs disruption (i.e., β -CD treatment) similarly affected the voltage sensor movement of both HCN2 and 4 isoforms, while SVD movement of HCN1 is insensitive to OMDs disruption. The authors, indeed, stated that "This reduction (FRET on β -CD treated cells expressing HCN2, Fig. 5) closely mirrored the changes in FRET between corresponding sites in HCN4 channels (Handlin et al., Nat Communication 2023). In contrast, minimal FRET change was detected between comparable sites in HCN1 channels following β -CD treatment, as shown previously (Handlin et al., Nat Communication 2023).

We thank the reviewer for the positive comments and careful review of our manuscript.

However, I found some weaknesses in the electrophysiology results that in my opinion are overinterpreted and, as they are, do not support the main claim. The evidence for HCN2 activation, due to an increase in open probability, is not convincing and requires some explanations but in my opinion also additional experiments.

Figure 5-a-c: It is surprising that β -CD does not shift the activation curve of HCN2, while it does (about +11 mV) in HCN4 (Barbuti et al, 2006). Authors need to address this difference.

There is subtype variability for the shift of the G-V curve caused by \$\beta\$ -CD. Indeed, \$\beta\$ -CD shifts the G-V curve for HCN4, which we also reported previously. Small DRG neurons have a mixture of HCN1, HCN2 and HCN3, although HCN2 is predominant, as evidenced by considerable cAMP sensitivity. Nevertheless, it's not an overexpression system with only one HCN subtype. In addition, there is greater shift in the G-V curve introduced by \$\beta\$ -CD for HCN4 than for HCN1, as previously reported (Fürst and D'Avanzo, Sci Rep. 2015; 5: 14270.; Handlin and Dai, Nat. Comms. 2023). The greater shift in HCN4 by \$\beta\$ -CD might be related to the inherent left-shifted activation curve of HCN4, compared to HCN1 and HCN2.

We conducted additional experiments by overexpressing human HCN2-YFP in HEK cells. These experiments revealed similarly little shift in the G-V curve by β -CD, but a noticeable change in the slope factor.

Figure 5C- the change in slope is truly minimal and can be due to many other experimental problems rather than being a real effect. For instance, as one can judge from the very few original data shown, the one in Figure 5a shows that the voltage protocol is not adequate to properly record HCN2 current as it does not allow to reach full activation state of the current at depolarized voltages. The statistics of Figure 2E shows that the difference, converted in apparent gating charge, is barely significant. It seems odd to draw all the conclusions on the increase in P_o at -60 mV from this puzzling set of results. Increasing n , the number of recordings, is definitively needed, as well as improving the protocol, as mentioned above, and most important, showing the original data of the currents. It would also help to include a table with individual $V_{1/2}$ and slope values from the Boltzmann fitting.

In the revised version, we reinforced this conclusion, as the reviewer suggested, by increasing the number of recordings, displaying original data, adding a new table 1 in the supplemental information, performing same-patch β -CD application experiment, and incorporating an extended 5-second-long voltage protocol.

In our resubmission, we included new data (Fig. 5) where β -CD was applied to the same patch, allowing us to compare channel current amplitude before and after application. It is not feasible to study P_o using whole-cell patch-clamp recordings across different cells. In the same-patch experiments, we observed a significant increase in current amplitude, further demonstrating that overall P_o increases following the disruption of OMDs.

We acknowledge that the 2-second voltage protocol is relatively short, especially at less hyperpolarized voltages. Shorter protocols tend to emphasize the kinetic aspects of channel activation, whereas longer protocols allow proteins to equilibrate and reach thermodynamic stability. While the change in the slope of the activation curve using the 2-second short protocol may seem less dramatic, it is highly reproducible, observed in both in vitro paclitaxel-induced pain models and in vivo SNI pain models. Nonetheless, when we extended the duration to 5 seconds, which generated slightly steeper slope of the G-V relationship (and higher apparent gating charge, see Fig. 5 and Supplementary Table 1), we observed similar effects with both β -CD and paclitaxel compared to using the shorter voltage protocol. For this study, we conclude that either protocol is acceptable, provided that all comparisons use the same hyperpolarization duration.

Figure 5e-

-The gating charge value seems low. Is this comparable with other z values reported in the literature?

The gating charge value is lower than those reported in overexpression systems, such as in HEK cells or oocytes, and especially when using the shorter hyperpolarizing voltage duration to activate the channel. Longer voltage protocol provided slightly higher gating charge value. Measuring a mixed population of HCN1, HCN2, and HCN3 channels in nociceptor DRG neurons could have contributed to this as well. This underestimation is why we refer to it as "apparent" gating charge. It is important to clarify that the primary focus of this paper is not on quantifying the absolute gating charge movement of HCN, but on the directional change in the gating charge.

To avoid this confusion, in the revised version, we have chosen to present the slope factor of the G-V curve directly in the figures instead of the apparent gating charge.

- The gating charge does not change with cAMP. This is in contrast with other approaches showing that it changes, (Hummert et al, 2018 DOI: 10.1371/journal.pcbi.1006045) and should at least be commented on.

We found, in the presence cAMP, the slope of the G-V curve is slightly decreased (higher estimated gating charge) for native HCN channels in rat DRG neurons. Indeed, this is opposite to the Hummert et al., 2018 paper using mouse HCN2 in oocytes. In some previous papers, the slope factor for mouse HCN2 barely changed after cAMP (including Chen et al., 2001, J Gen Physiol (2001) 117 (5): 491–504). However, another previous study rigorously characterized the cAMP-induced change in the slope factor of G-V fitting for mHCN2, showing a similar decrease to that observed in our experiments (Kusch et al., Neuron, 2010, DOI: 10.1016/j.neuron.2010.05.022).

We have added a comment on this point in the paper: “The cAMP-induced decrease in the slope factor of the G-V relationships is subtle but noticeable from native HCN channels of DRG neurons. This finding aligns with previous research showing that cAMP similarly decreased the slope factor in mouse HCN2 (Kusch et al., 2010). However, it contrasts with a previous study suggesting that cAMP binding decreases the gating charge (Hummert et al., 2018).”

In summary, we find that the cAMP-induced increase in gating charge during HCN channel activation in rat DRG neurons is small but real. While this effect is an intriguing observation, it is not central to the conclusions of this paper.

-It would be important to show raw data for all treatments, given the small effect we are dealing with. Authors should show exemplary currents recorded after the treatment with β -CD, paclitaxel, with and without cAMP.

Done, added to Fig. 5.

Figure 5f

- shows that ΔG is the same without and with cAMP. How can it be? cAMP binding provides the energy to facilitate voltage gating, and ΔG should be reduced.

The Authors use the following formula to calculate $\Delta G = RT V_{1/2}/V_s$
with $V_{1/2}$, half activation voltage
 V_s , the slope factor,
RT are constant

Since $V_{1/2}$ changes, ΔG cannot be the same in both cases.

Using this formula (Li-Smerin, Hackos, Swartz, JGP, 2000, Dai and Zagotta, eLife, 2017), the ΔG is affected by both $V_{1/2}$ and the slope factor V_s . We found that in the presence of cAMP, there is a less negative $V_{1/2}$, but also a slightly decreased V_s , which tends to attenuate the effect of $V_{1/2}$ on ΔG . Using this formula, V_s contributes to the ΔG as much as the $V_{1/2}$. This formula provides a suitable approach to derive ΔG_0 values from steady-state ionic current data in this paper.

For the isolated cyclic nucleotide-binding domain (CNBD), cAMP significantly reduces the free energy required for binding, as shown in multiple studies using soluble CNBD (Deberg et al., JCB, 2016; Eggan et al., eLife, 2024). Thus, cAMP is unequivocally an agonist for the CNBD. However, the favorable change in free energy induced by cAMP in the context of the full-length channel within a native environment is less pronounced. In our study, we found that the cAMP-induced change in ΔG for native HCN currents of small DRG neurons is noticeable especially when using the short

hyperpolarization protocol, though still not particularly pronounced. This ΔG change by cAMP is minimal using the long hyperpolarization protocol, because the change in the slope factor by cAMP attenuates the effect of $V_{1/2}$ on ΔG . Interpreting a small change in ΔG in terms of its physiological impact can be challenging, but the shift in the G-V curve by cAMP for HCN2 is clearly physiologically consequential.

It would be also helpful to extend the analysis of ΔG of activations also to HCN1 and HCN4. Indeed, ΔG values are used in the present work to corroborate the assumption that there is an OMD-dependent movement of HCN2 VSD. Therefore, a comparison of ΔG of activations among the three HCN subtypes will strengthen the conclusions of the work. To do not burden the main figures, I suggest to add a Suppl. Figure with ΔG of activations obtained from the analysis of HCN4 and HCN1 activation curves.

Added the ΔG information for HCN1, HCN2, and HCN4 in the Suppl. Fig. 11.

Minor changes:

-Please, use isotypes instead of isoforms.

Corrected.

-Please, organize Suppl. Figures based on their appearance in the text.

Checked and reorganized.

-Suppl. Fig. 5b and c. Statistics is absent, though significance of the treatments is highlighted in the text. Please, add statistics, and, if there is not significance, modify the text accordingly. Representative firings in panel A seems to indicate a difference. In case this is not statistically significant, please, explain why.

We have added statistical analysis to this section and moved the current injection-induced action potential firing from the supplementary information to the main Figure 3. We combined spontaneous and nonspontaneous firing in Figures 3d and 3e to provide a more comprehensive view of how the treatments affect neuronal excitability.

-From the electrophysiological point of view, there is clear difference between small and large DRG, as the first express a slow and cAMP-sensitive current, while the second a fast and insensitive one. In explaining this difference, I suggest to refer to the seminal papers discriminating between small and large DRGs on the basis of I_h properties. Nonetheless, it may be worth discriminating small DRGs also for their response to capsaicin (TRPV1), or perform some staining with nociceptive-small DRG markers (Nav1.8, for instance).

We have referred to the papers discriminating between small and large DRGs based on I_h properties. We have only considered small DRG neurons that have capacitance < 28 pF as nociceptive neurons for all related conditions. We have added the statement in the paper with reference cited: "Small nociceptor DRG neurons exhibit slow, cAMP-sensitive HCN currents whereas larger DRG neurons exhibit fast, but relatively cAMP-insensitive currents^{18,22}". In addition, a recent paper (Zheng, Y. et al. Neuron 103, 598-616 e597, 2019) showed that Nav1.8 and TRPV1 are also abundantly expressed in some (C-fiber) low-threshold mechanoreceptor DRG neurons. In addition, staining would interfere with some of our FLIM fluorescence experiments, so we decide not to use the antibody-based staining.

Reviewer #2 (Remarks to the Author):

In this manuscript, authors investigate the role of OMDs on neuronal response and protein function.

While the results are interesting, they are always in one direction, i.e., domains get smaller (by CD treatment) and resulting response is due to this change. It is necessary to show that the other way around is also possible, i.e., if you can make the OMDs bigger, the response will be the opposite. This could be done by inserting cholesterol in the membrane. Moreover, the impact of cholesterol specifically vs OMDs generally should be dissected by changing the OMD size without changing the cholesterol content. My major comments are below.

We thank the reviewer for insightful inquiries. We focused on the effect of decreasing OMD domains on neuronal response and channel function because the OMD size in normal nociceptor DRG neurons is already large, and our FRET probe data (newly added in Fig. 3) showed that treatment with water-soluble cholesterol (WSC) does not further increase OMD size in regular DRG neurons. Decrease in OMD sizes is also the direction of OMD change induced by neuropathic pain.

We agree that exploring mechanisms in the opposite direction may provide valuable therapeutic insights. In response, we conducted additional experiments, now presented in Fig. 8, showing that WSC treatment effectively restores OMD size in ipsilateral neurons of the SNI model. This restoration is associated with reduced neuronal firing, suggesting that the expansion of OMD could potentially alleviate pain.

We also explained the potential dual effects of cholesterol on HCN channels in the text of this section. "Overall, these findings suggest that cholesterol enrichment at the membrane has two distinct effects: increasing OMD size and directly binding to the channels. An increase in OMD size likely modifies membrane thickness and alters the movement of voltage sensors, leading to a reduction in the apparent gating charge movement. Direct cholesterol binding may become functionally significant when the OMD size is sufficiently large to accommodate most of the HCN channels into OMDs. In this case, further increasing cholesterol levels causes a shift in the G-V relationship without affecting its slope, likely by altering the coupling between the voltage sensor and the channel pore. Further research is needed to fully understand how direct cholesterol binding influences the electromechanical coupling of HCN channels."

We acknowledge that dissecting the effects of cholesterol versus OMDs presents an important future direction, and a technical challenge, given that cholesterol is an inherent component of OMDs. There are limited ways to physiologically change the OMD size without changing the cholesterol level. Low concentrations of SDS would do it, but since it's a detergent, it might produce unwanted effects to channel gating. Studying cholesterol regulation of purified HCN channels in a reductionist, i.e. proteoliposome, system might overlook the influence of other lipid components present in a native membrane and might not be able to mimic the lipid domain separation in native membranes. Recognizing this limitation, we have added a paragraph to the end of the discussion to address this specific issue. Nevertheless, this is beyond the scope of this paper and will be a future direction.

Fig 1: The observed effects can be due to the access membrane and resulting topology. To rule this out, authors should perform osmotic swelling on the cells which should not change the results if the observed effects are due to lipid domains.

Some images should be show in the main Figures to judge the quality of the images, staining etc.

We have performed the swelling treatment by applying hypotonic incubation of cells, and found the effects by β -CD, WSC and low concentration of detergents were maintained (Supplementary Fig. 2g). Thus, we think it's less likely the observed effects were due to changes in access membrane. Our directional change in FRET using L10-CFP/L10-YFP probes (increase by β -CD and decrease by WSC) is consistent with a previous paper from Bertil Hille's lab using the same pair of probes (Myeong et al., PNAS, 2021). This 2021 paper particularly demonstrated that the effects of β -CD and WSC were due to

a change in the fraction of ordered versus disordered membrane phases. We were able to reproduce the FRET results using the same probes. In addition, we have added representative images to Fig. 1, highlighting the FRET changes in tsA cells.

Figure 2: would the cell type differences be due to differential GM1 abundance? How does the brightness of the staining compare in different cell types and is there any correlation between staining and FRET efficiency?

Proper statistical tests should be applied to the data.

We thank the reviewer for raising insightful points, particularly regarding the dependence of our FLIM-FRET approach on cholera toxin and its interaction with GM1 gangliosides. We acknowledge the importance of GM1 in OMDs and the fact that its abundance can vary among different cell types. Thus, in addition to CTxB as probes, we used the genetically encoded L10-CFP and L10-YFP probes, which are independent of the GM1 abundance, as another probe for quantifying the localization of ion channels in proximity to OMDs. This result is consistent with the FRET data based on the cholera toxin subunit B probes. It provides another piece of evidence supporting our conclusions.

In physiological conditions, GM1 constitutes a small percentage of total membrane lipids. The nervous system features the highest proportion; on average, GM1 accounts for 1% of all lipids (Holthuis et al., *Physiological Reviews*, 2001; Sipione et al., *Frontiers in Neuroscience*, 2020). While GM1 levels may differ, it is not self-contradictory to state that cells with larger OMDs could inherently contain higher concentrations of GM1. This correlation was suggested by a prior study (Shi et al., *JACS*, 2007), which demonstrated varying ordered domain diameters based on GM1 percentages (13.6 nm for 1%, 15.3 nm for 3%, and 18.6 nm for 5%). Furthermore, the cholera toxin subunit B (CTxB) functions as a pentameric multi-valent ligand, necessitating multiple GM1 bindings for interaction with the plasma membrane. This characteristic aligns with the close proximity distribution of multiple GM1, a hallmark of OMDs. CTxB binding requires sufficient GM1, and even at 1%, GM1 has been shown to facilitate efficient CTxB binding. Conversely, an excessive concentration of GM1 leads to clustering, inhibiting CTxB binding (Shi et al., *JACS*, 2007, Šachl et al., *BBA*, 2015). An excess of GM1 does not further augment CTxB binding. Thus, the physiological variability in GM1 may not significantly impact our FRET assay using CTxB.

Our fluorescence lifetime-based imaging technique offers advantages over traditional fluorescence intensity-based imaging by being largely independent of factors such as fluorescence brightness, photobleaching, and autofluorescence. We measured FRET efficiency by comparing the donor lifetime alone with the donor lifetime in the presence of the acceptor. The measured FRET efficiency is mostly unaffected by variations in the number of transfected probes. For example, when comparing small and large DRG neurons, we analyzed CTxB-AF488 brightness and FRET efficiency. In one dataset collected on the same day, and with same imaging conditions, a significant difference in FRET efficiency was observed between small and large DRG neurons (** $p = 0.0015$), while the donor brightness difference was not significant ($p = 0.11$). Linear regression analysis showed little to no correlation between brightness and FRET efficiency. While brightness primarily reflects the extent of CTxB binding, FRET efficiency offers insights into the relative size of the OMD.

In the revised version, we have added statistics.

Figure 3: Authors should show that the opposite of this scenario is also possible: i.e., when OMDs are enhanced, the firing frequency gets smaller. Otherwise, the observed effect can be due to cholesterol but not generally OMDs.

Consistent with the idea of expanding OMDs, we observed that supplementing neurons with cholesterol reduced spontaneous firing in small DRG neurons (see new Fig. 3). Additionally, applying WSC treatment to the ipsilateral side of SNI pain model neurons effectively enhanced OMDs, resulting in a decreased firing frequency (new Fig. 8). This effect likely stems from changes in OMDs rather than shifts in the G-V curve, as cholesterol primarily affects the activation curve without altering its slope. In contrast, WSC application to ipsilateral neurons did not shift the G-V curve but did decrease the slope factor (indicating increased apparent gating charge movement), which aligns with the notion of OMD expansion. A cartoon illustrating these dual effects of cholesterol on HCN channels is also provided in Fig. 8.

Figure 4: Why isn't there the data from Lck construct? It is really helpful to show both of them in parallel to increase the robustness of the results.

We have added new data to the figure 4 using the L10-CFP/L10-YFP FRET method, showing increased FRET with paclitaxel treatment. This directional change aligns with the properties of the L10 probes, and contrasts with the FRET changes observed using the CTxB probe method, which is consistent with a decrease in OMD sizes. In general, the CTxB method is preferable for primary neurons because it doesn't require lipofectamine (small DRG) or electroporation (large DRG)-mediated transfection of L10 probes. L10 probes were used to validate the CTxB probe method, as shown in Figures 1-3.

Figure 5: How can HCN2 change its domain localization. Based on recent papers, the localization of a protein in OMDs are determined by the protein structure, post-translational modifications etc. How does a protein leave the OMDs? This requires an explanation. Again, the opposite experiment (OMD enhancement) will be useful.

To clarify this point, we have modified the schematic cartoon in the Fig. 7 (formerly Fig. 6) and one sentence in the abstract, which was misleading in the previous version of the manuscript. In neuropathic pain conditions, we think the ability of HCN2 channels to incorporate into OMDs remains unchanged. However, the disruption of OMDs results in HCN channels being more exposed to a disordered lipid environment. In other words, there are not sufficient OMDs to accommodate HCN channels.

Reviewer #3 (Remarks to the Author):

I co-reviewed this manuscript with one of the reviewers who provided the listed reports. This is part of the Nature Communications initiative to facilitate training in peer review and to provide appropriate recognition for Early Career Researchers who co-review manuscripts

See the response to other reviewers.

Reviewer #1 (Remarks to the Author):

The Authors have replied to all the points raised, and cleared most of them. There is just one answer that is not completed though. When I was asking for deltaG values for HCN1 and HCN2, I meant before and after the addition of beta-CD, while the Authors in their reply have added DeltaG values in the absence of beta-CD only.

According to the hypothesis presented by the Authors after the treatment with β -CD deltaG values become more positive in HCN2 and HCN4 (because their VSD are sensitive to cholesterol) and are unchanged in HCN1, that is cholesterol insensitive. This is while it would be important to know the values.

[Also note that the values were added to the legend of Suppl Figure 10, and not of Suppl. Figure 11, as mentioned by the Authors].

In the final submission, we have added the 'change in ΔG ' ($\Delta\Delta G$) due to β -CD application, $\Delta\Delta G = \Delta G_{\beta\text{-CD}} - \Delta G_{\text{Control}}$ for the three HCN subtypes. As the reviewer pointed out, the $\Delta\Delta G$ is much greater for HCN2 (1.77 kcal/mol) and HCN4 (2.58 kcal/mol) than for HCN1 (0.70 kcal/mol).

Yes, this is added in the legend of Suppl Figure 10, not in Suppl. Figure 11.

Reviewer #2 (Remarks to the Author):

Authors addressed my major concerns. Some points, such as direct role of cholesterol vs OMDs, were not directly addressed, however, the authors discussed these limitations in the manuscript. Therefore, I am satisfied with the revised version.

Thank you.

Reviewer #3 (Remarks to the Author):
